# Succinate mediates inflammation-induced adrenocortical dysfunction

Ivona Mateska[1]*, Anke Witt[1], Eman Hagag[1], Anupam Sinha[1], Canelif Yilmaz[1], Evangelia Thanou[2], Na Sun[3], Ourania Kolliniati[4], Maria Patschin[1], Heba Abdelmegeed[1], Holger Henneicke[5,6], Waldemar Kanczkowski[1], Ben Wielockx[1], Christos Tsatsanis[4], Andreas Dahl[7], Axel Karl Walch[3], Ka Wan Li[2], Mirko Peitzsch[1], Triantafyllos Chavakis[1], Vasileia Ismini Alexaki[1]*

[1]Institute of Clinical Chemistry and Laboratory Medicine, University Hospital, Technische Universität Dresden, Dresden, Germany; [2]Center of Neurogenomics and Cognitive Research (CNCR), Department of Molecular and 10 Cellular Neurobiology, Vrije Universiteit, Amsterdam, Netherlands; [3]Research Unit Analytical Pathology, German Research Center for Environmental Health, Helmholtz Zentrum München, Munich, Germany; [4]Department of Clinical Chemistry, Medical School, University of Crete, Heraklion, Greece; [5]Department of Medicine III & Center for Healthy Ageing, Technische Universität Dresden, Dresden, Germany; [6]Center for Regenerative Therapies, TU Dresden, Technische Universität Dresden, Dresden, Germany; [7]DRESDEN-concept Genome Center, Center for Molecular and Cellular Bioengineering, Technische Universität Dresden, Dresden, Germany

*For correspondence:
Ivona.Mateska@uniklinikum-dresden.de (IM);
VasileiaIsmini.Alexaki@uniklinikum-dresden.de (VIsminiA)

Competing interest: The authors declare that no competing interests exist.

**Abstract** The hypothalamus-pituitary-adrenal (HPA) axis is activated in response to inflammation leading to increased production of anti-inflammatory glucocorticoids by the adrenal cortex, thereby representing an endogenous feedback loop. However, severe inflammation reduces the responsiveness of the adrenal gland to adrenocorticotropic hormone (ACTH), although the underlying mechanisms are poorly understood. Here, we show by transcriptomic, proteomic, and metabolomic analyses that LPS-induced systemic inflammation triggers profound metabolic changes in steroidogenic adrenocortical cells, including downregulation of the TCA cycle and oxidative phosphorylation, in mice. Inflammation disrupts the TCA cycle at the level of succinate dehydrogenase (SDH), leading to succinate accumulation and disturbed steroidogenesis. Mechanistically, IL-1β reduces SDHB expression through upregulation of DNA methyltransferase 1 (DNMT1) and methylation of the *SDHB* promoter. Consequently, increased succinate levels impair oxidative phosphorylation and ATP synthesis and enhance ROS production, leading to reduced steroidogenesis. Together, we demonstrate that the IL-1β-DNMT1-SDHB-succinate axis disrupts steroidogenesis. Our findings not only provide a mechanistic explanation for adrenal dysfunction in severe inflammation, but also offer a potential target for therapeutic intervention.

## Editor's evaluation

Acute inflammation in mammals activates the hypothalamic pituatary axis leading to increased glucocorticoid release, which is required to restrain the inflammatory response. However, in settings of severe or prolonged inflammation, such as that seen in sepsis, there is reduced adrenal steridogenesis. The studies described in this paper provide a plausible mechanism for adrenal resistance which develops during excessive inflammation. The revisions have improved the paper and the Methods are sound.

## Introduction

Stress triggers the hypothalamic-pituitary-adrenal (HPA) axis, that is, the release of corticotropin-releasing hormone from the hypothalamus, followed by adrenocorticotropic hormone (ACTH) secretion from the anterior pituitary, which stimulates the synthesis of glucocorticoid hormones in the adrenal cortex, primarily cortisol in humans and corticosterone in rodents (*Chrousos, 1995*; *Lightman et al., 2020*; *Payne and Hales, 2004*). Similar to any other stress stimulus, inflammation activates the HPA axis leading to increased glucocorticoid release, which is required to restrain the inflammatory response (*Alexaki, 2021a*; *Alexaki and Henneicke, 2021b*; *Kanczkowski et al., 2013a*; *Kanczkowski et al., 2013b*; *Kanczkowski et al., 2013c*). Adrenalectomized rodents show increased mortality after induction of systemic inflammation, while glucocorticoid administration increases survival (*Bertini et al., 1988*; *Butler et al., 1989*). Essentially, severe inflammation in sepsis is associated with impaired adrenal gland function (*Annane et al., 2000*; *Boonen et al., 2015*; *Boonen et al., 2014*; *den Brinker et al., 2005*; *Jennewein et al., 2016*), but the mechanisms remain poorly understood.

In immune cells, such as macrophages, dendritic cells, and T cells, inflammation triggers cellular metabolic reprograming, enabling the cells to meet the increased demands for fast energy supply and anabolic processes (*Geltink et al., 2018*; *O'Neill and Pearce, 2016*; *Ryan and O'Neill, 2020*). How inflammation may affect cellular metabolism in other cell types and how this affects their function is less explored. Here, we show that LPS-induced inflammation profoundly changes the cellular metabolism of steroidogenic adrenocortical cells, perturbing the TCA cycle at the level of succinate dehydrogenase B (SDHB). This is coupled to succinate accumulation, which impairs oxidative phosphorylation and leads to reduced steroidogenesis. Mechanistically, IL-1β inhibits *SDHB* expression through DNA methyltransferase 1 (DNMT1)-dependent DNA methylation of the *SDHB* promoter.

## Results

### Metabolic reprograming of the adrenal cortex in inflammation

To explore inflammation-induced alterations in the adrenal cortex, we performed RNA-Seq in microdissected adrenal cortices from mice treated for 6 hr i.p. with 1 mg/kg LPS or PBS, which revealed 2,609 differentially expressed genes, out of which 1,363 were down- and 1,246 were upregulated (*Figure 1A*). Gene set enrichment analysis (GSEA) using the Molecular Signatures Database (MSigDB) hallmark gene set collection (*Liberzon et al., 2015*) showed a significant enrichment of inflammatory response-related gene sets in the adrenal cortex of LPS-treated mice (*Figure 1B*). In acute inflammation, leukocytes infiltrate the adrenal cortex (*Kanczkowski et al., 2013b*) and resident macrophages are activated (*González-Hernández et al., 1994*; *Schober et al., 1998*). In order to delineate the inflammatory response in the adrenocortical steroidogenic cells, CD31⁻CD45⁻ cells were sorted: enrichment in steroidogenic cells was evidenced by high steroidogenic acute regulatory protein (*Star*) expression (*Figure 1—figure supplement 1a*), and purity was verified by the absence of *Cd31* and *Cd45* expression (*Figure 1—figure supplement 1B and C*). Moreover, we confirmed the absence of expression of the medullar markers tyrosine hydroxylase (*Th*) and phenylethanolamine *N*-methyltransferase (*Pnmt*) in isolated cortices and adrenocortical steroidogenic cells (*Figure 1—figure supplement 1D and E*). Proteomic analysis in the sorted CD31⁻CD45⁻ adrenocortical cell population and GSEA of GO terms confirmed the enrichments of innate immune response-related proteins in adrenocortical cells of LPS-injected mice (*Figure 1C*), suggesting that steroidogenic adrenocortical cells respond to inflammatory stimuli.

LPS treatment leads to increased plasma corticosterone levels (*Kanczkowski et al., 2013a*; *Kanczkowski et al., 2013b*; *Kanczkowski et al., 2013c*). Numerous studies have shown that elevated glucocorticoid levels are primarily driven by activation of the HPA axis and coincide with increased circulating ACTH levels (*Kanczkowski et al., 2013a*; *Kanczkowski et al., 2013b*; *Kanczkowski et al., 2013c*). This is accompanied by increased expression of genes related to steroid biosynthesis (*Chen et al., 2019b*). We confirmed increased expression of the cholesterol transporter *Star* (*Miller, 2007*) and the terminal enzyme for glucocorticoid synthesis *Cyp11b1* (*Payne and Hales, 2004*) in adrenocortical cells of LPS mice (*Figure 1—figure supplement 2A and B*). However, the expression of genes encoding for other steroidogenic enzymes, such as 3β-hydroxysteroid dehydrogenase 2 (*Hsd3b2*) and *Cyp21a1*, was reduced, while *Cyp11a1* remained unchanged (*Figure 1—figure supplement 2C–E*). Similarly, protein levels of steroidogenic factor 1 (SF-1), a key inducer of steroidogenesis (*Parker,*

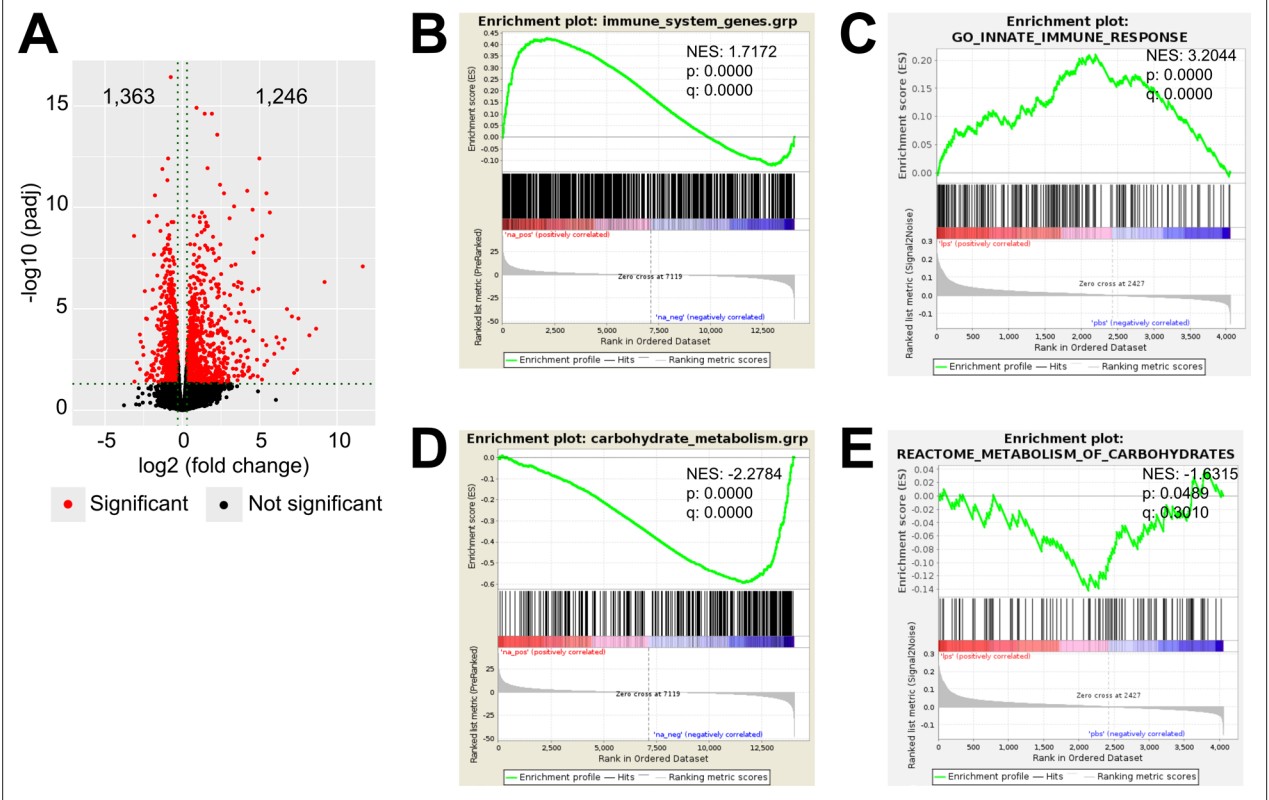

**Figure 1.** LPS-induced inflammation changes the transcriptional and proteomic profile of the adrenal cortex. (**A**) Volcano plot showing differentially expressed genes in the microdissected adrenal gland cortex of mice treated for 6 hr with PBS or LPS. (**B**) Gene set enrichment analysis (GSEA) for immune pathways in the adrenal cortex of LPS versus PBS mice. (**C**) GSEA for proteins associated with the innate immune response in CD31⁻CD45⁻ adrenocortical cells of mice treated for 24 hr with PBS or LPS. (**D**) RNA-Seq-based GSEA for carbohydrate metabolism in the adrenal cortex of LPS versus PBS mice. (**E**) GSEA for proteins associated with carbohydrate metabolism in CD31⁻CD45⁻ adrenocortical cells of LPS versus PBS mice. NES: normalized enrichment score. (**A,B,D**) n=3 mice per group, (**C,E**) n=6 mice per group, padj <0.05 was used as a cut-off for significance.

The online version of this article includes the following source data and figure supplement(s) for figure 1:

**Source data 1.** LPS-induced inflammation changes the transcriptional and proteomic profile of the adrenal cortex.

**Figure supplement 1.** Efficiency of CD31⁻CD45⁻, immune (CD45⁺), and endothelial (CD31⁺) cell sorting.

**Figure supplement 1—source data 1.** Efficiency of CD31-CD45-, immune (CD45⁺), and endothelial (CD31⁺) cell sorting.

**Figure supplement 2.** Inflammation-associated changes in the steroidogenic pathway.

**Figure supplement 2—source data 1.** Inflammation-associated changes in the steroidogenic pathway.

*1999*), were somewhat reduced after LPS injection (*Figure 1—figure supplement 2F*). Therefore, the observed changes in plasma glucocorticoid levels which accompany inflammation cannot be solely explained by the transcriptional changes in steroidogenic enzymes.

Next, we explored the cell metabolic changes induced by LPS in the adrenal cortex. By GSEA of the RNA-Seq data, we observed negative regulation of gene sets related to carbohydrate metabolism in the adrenal cortex of LPS-injected mice (*Figure 1D*). Proteomic analysis was performed in CD31⁻CD45⁻ adrenocortical cells (*Figure 1—figure supplement 1*) to examine the effects of inflammation specifically on the metabolism of steroidogenic adrenocortical cells, evading the well-described inflammation-induced metabolic changes in immune cells (*Geltink et al., 2018*; *O'Neill and Pearce, 2016*; *Ryan and O'Neill, 2020*). Similarly to the RNA-Seq data, GSEA of the proteomic data showed significant negative enrichment of proteins associated with carbohydrate metabolism in the steroidogenic cells (*Figure 1E*). EGSEA pathway analysis of the RNA-Seq and proteomic data revealed that TCA cycle, oxidative phosphorylation, tyrosine metabolism, fatty acid degradation, D-glutamine and D-glutamate metabolism, glutathione metabolism, and other metabolic pathways were significantly enriched among the downregulated genes and proteins in the adrenal cortex and in steroidogenic cells of LPS mice (*Table 1*, *Table 2*).

**Table 1.** Cellular metabolic pathways transcriptionally regulated by inflammation in the adrenal cortex. The pathway analysis of differentially expressed genes was done with the software package EGSEA and queried against the KEGG pathways repository. Pathways with p<0.05 are shown.

| ID | Metabolic pathway | Number of expressed genes | p-Value | padj | avg.logfc | Direction |
|---|---|---|---|---|---|---|
| mmu00190 | Oxidative phosphorylation | 132/134 | 8.32E-15 | 7.32E-13 | 0.613034377 | Down |
| mmu00280 | Valine, leucine, and isoleucine degradation | 55/56 | 1.16E-05 | 0.001119154 | 0.725758563 | Down |
| mmu00511 | Other glycan degradation | 18/18 | 3.82E-05 | 0.001119154 | 0.29921956 | Down |
| mmu00980 | Metabolism of xenobiotics by cytochrome P450 | 65/65 | 0.000282456 | 0.006214029 | 0.650304677 | Down |
| mmu00350 | Tyrosine metabolism | 38/39 | 0.001754788 | 0.030884268 | 0.305935415 | Down |
| mmu00640 | Propanoate metabolism | 31/31 | 0.002151447 | 0.031554563 | 0.886410728 | Down |
| mmu00020 | Citrate cycle (TCA cycle) | 32/32 | 0.003091828 | 0.034636464 | 0.21912487 | Down |
| mmu01200 | Carbon metabolism | 118/118 | 0.003685749 | 0.034636464 | 0.715139933 | Down |
| mmu00471 | D-Glutamine and D-glutamate metabolism | 3/3 | 0.003980826 | 0.034636464 | 0.338725016 | Up |
| mmu00300 | Lysine biosynthesis | 2/2 | 0.004518807 | 0.034636464 | 0.156233777 | Down |
| mmu00630 | Glyoxylate and dicarboxylate metabolism | 29/29 | 0.006079126 | 0.034636464 | 0.886410728 | Down |
| mmu00071 | Fatty acid degradation | 49/49 | 0.006124673 | 0.034636464 | 0.401441917 | Down |
| mmu01210 | 2-Oxocarboxylic acid metabolism | 19/19 | 0.006297539 | 0.034636464 | 0.942661129 | Down |
| mmu00920 | Sulfur metabolism | 11/11 | 0.006297539 | 0.034636464 | 0.601864913 | Down |
| mmu00480 | Glutathione metabolism | 58/59 | 0.006297539 | 0.034636464 | 0.550257753 | Down |
| mmu00510 | N-Glycan biosynthesis | 49/49 | 0.006297539 | 0.034636464 | 0.254426968 | Down |
| mmu00450 | Selenocompound metabolism | 17/17 | 0.009115527 | 0.047186259 | 0.329118527 | Up |
| mmu00514 | Other types of O-glycan biosynthesis | 22/22 | 0.010614958 | 0.050693394 | 0.204924928 | Up |
| mmu00440 | Phosphonate and phosphinate metabolism | 6/6 | 0.010945165 | 0.050693394 | 0.291510253 | Up |
| mmu00120 | Primary bile acid biosynthesis | 16/16 | 0.01193422 | 0.052510566 | 2.713591551 | Down |
| mmu00565 | Ether lipid metabolism | 44/44 | 0.016711812 | 0.061724814 | 0.433837026 | Down |
| mmu00520 | Amino sugar and nucleotide sugar metabolism | 49/49 | 0.018212661 | 0.061724814 | 0.652646241 | Down |
| mmu00790 | Folate biosynthesis | 14/14 | 0.01821569 | 0.061724814 | 0.420000237 | Down |
| mmu00230 | Purine metabolism | 174/178 | 0.018793387 | 0.061724814 | 0.830115541 | Down |
| mmu00603 | Glycosphingolipid biosynthesis – globo series | 16/16 | 0.019832034 | 0.061724814 | 0.534287429 | Up |
| mmu00534 | Glycosaminoglycan biosynthesis – heparan sulfate/heparin | 24/24 | 0.020491524 | 0.061724814 | 0.471378176 | Up |
| mmu00270 | Cysteine and methionine metabolism | 46/48 | 0.020959507 | 0.061724814 | 0.601864913 | Down |
| mmu01100 | Metabolic pathways | 1303/1315 | 0.022419002 | 0.061724814 | 0.683947211 | Down |
| mmu00531 | Glycosaminoglycan degradation | 21/21 | 0.023139686 | 0.061724814 | 0.911753864 | Down |
| mmu00604 | Glycosphingolipid biosynthesis – ganglio series | 15/15 | 0.023268716 | 0.061724814 | 0.457285673 | Down |

*Table 1 continued on next page*

*Table 1 continued*

| ID | Metabolic pathway | Number of expressed genes | p-Value | padj | avg.logfc | Direction |
|---|---|---|---|---|---|---|
| mmu00250 | Alanine, aspartate, and glutamate metabolism | 36/37 | 0.023685519 | 0.061724814 | 0.942661129 | Down |
| mmu00240 | Pyrimidine metabolism | 101/104 | 0.024235887 | 0.061724814 | 0.308135641 | Up |
| mmu00130 | Ubiquinone and other terpenoid-quinone biosynthesis | 11/11 | 0.024630584 | 0.061724814 | 0.40991964 | Down |
| mmu00330 | Arginine and proline metabolism | 49/50 | 0.025088792 | 0.061724814 | 0.479554867 | Down |
| mmu00564 | Glycerophospholipid metabolism | 94/94 | 0.025448859 | 0.061724814 | 0.498146256 | Down |
| mmu00910 | Nitrogen metabolism | 17/17 | 0.025782601 | 0.061724814 | 1.765582115 | Up |
| mmu00982 | Drug metabolism – cytochrome P450 | 67/67 | 0.027201304 | 0.061724814 | 0.620562175 | Down |
| mmu00785 | Lipoic acid metabolism | 3/3 | 0.027507553 | 0.061724814 | 0.133403173 | Down |
| mmu00051 | Fructose and mannose metabolism | 35/35 | 0.029679889 | 0.061724814 | 0.652646241 | Down |
| mmu00561 | Glycerolipid metabolism | 59/59 | 0.030466773 | 0.061724814 | 0.4155946 | Down |
| mmu00512 | Mucin type O-glycan biosynthesis | 28/28 | 0.030540327 | 0.061724814 | 0.472458211 | Up |
| mmu00052 | Galactose metabolism | 32/32 | 0.031376559 | 0.061724814 | 0.42838689 | Down |
| mmu01230 | Biosynthesis of amino acids | 78/78 | 0.031800904 | 0.061724814 | 0.942661129 | Down |
| mmu00533 | Glycosaminoglycan biosynthesis – keratan sulfate | 14/14 | 0.031872109 | 0.061724814 | 0.27544313 | Down |
| mmu00900 | Terpenoid backbone biosynthesis | 22/23 | 0.032223321 | 0.061724814 | 0.320850492 | Up |
| mmu00730 | Thiamine metabolism | 15/15 | 0.032265244 | 0.061724814 | 0.323024942 | Down |
| mmu00500 | Starch and sucrose metabolism | 33/33 | 0.033413545 | 0.062178073 | 0.452533726 | Up |
| mmu00770 | Pantothenate and CoA biosynthesis | 18/18 | 0.033915312 | 0.062178073 | 0.307590054 | Down |
| mmu00062 | Fatty acid elongation | 27/27 | 0.035703854 | 0.062200773 | 0.536297353 | Down |
| mmu00592 | Alpha-linolenic acid metabolism | 25/25 | 0.035861114 | 0.062200773 | 0.568488547 | Down |
| mmu00562 | Inositol phosphate metabolism | 70/70 | 0.037060597 | 0.062200773 | 0.19317479 | Up |
| mmu00760 | Nicotinate and nicotinamide metabolism | 34/35 | 0.037777703 | 0.062200773 | 0.44361797 | Down |
| mmu00740 | Riboflavin metabolism | 8/8 | 0.038531607 | 0.062200773 | 0.336638606 | Down |
| mmu00670 | One carbon pool by folate | 19/19 | 0.039807001 | 0.062200773 | 0.293141874 | Up |
| mmu00310 | Lysine degradation | 57/59 | 0.040221284 | 0.062200773 | 0.409052131 | Up |
| mmu00010 | Glycolysis/gluconeogenesis | 66/66 | 0.040372513 | 0.062200773 | 0.4474754 | Down |
| mmu00053 | Ascorbate and aldarate metabolism | 27/27 | 0.040535303 | 0.062200773 | 0.319047565 | Down |
| mmu00030 | Pentose phosphate pathway | 32/32 | 0.040995964 | 0.062200773 | 0.316347941 | Down |

**Table 2.** Cellular metabolic pathways regulated on protein level by inflammation in adrenocortical cells. The pathway analysis of differentially expressed proteins was done with the software package EGSEA and queried against the KEGG pathways repository. Pathways with p<0.05 are shown.

| ID | Metabolic pathway | Number of detected proteins | p-Value | padj | avg.logfc | Direction |
|---|---|---|---|---|---|---|
| mmu00534 | Glycosaminoglycan biosynthesis – heparan sulfate/heparin | 4/24 | 1.50E-07 | 1.29E-05 | 0.13 | Up |
| mmu00280 | Valine, leucine, and isoleucine degradation | 37/57 | 5.20E-07 | 2.23E-05 | 0.01 | Down |
| mmu00983 | Drug metabolism – other enzymes | 25/92 | 3.80E-05 | 0.000856142 | 0.02 | Down |
| mmu00562 | Inositol phosphate metabolism | 19/72 | 5.89E-05 | 0.000856142 | 0.03 | Up |
| mmu00260 | Glycine, serine, and threonine metabolism | 19/40 | 6.89E-05 | 0.000856142 | 0.03 | Down |
| mmu00240 | Pyrimidine metabolism | 25/58 | 6.98E-05 | 0.000856142 | 0.04 | Up |
| mmu00982 | Drug metabolism – cytochrome P450 | 16/71 | 7.76E-05 | 0.000856142 | 0.02 | Down |
| mmu00230 | Purine metabolism | 44/133 | 7.96E-05 | 0.000856142 | 0.04 | Up |
| mmu00790 | Folate biosynthesis | 9/26 | 9.75E-05 | 0.000931452 | 0.09 | Up |
| mmu01100 | Metabolic pathways | 593/1608 | 0.000134694 | 0.00151176 | 0.03 | Down |
| mmu00190 | Oxidative phosphorylation | 70/135 | 0.000160368 | 0.00151176 | 0.02 | Down |
| mmu00980 | Metabolism of xenobiotics by cytochrome P450 | 19/73 | 0.000175997 | 0.00151176 | 0.02 | Down |
| mmu01240 | Biosynthesis of cofactors | 74/154 | 0.000178113 | 0.00151176 | 0.04 | Down |
| mmu00730 | Thiamine metabolism | 5/15 | 0.000187401 | 0.00151176 | 0.02 | Down |
| mmu00020 | Citrate cycle (TCA cycle) | 30/32 | 0.00020177 | 0.001156812 | 0.01 | Down |
| mmu01200 | Carbon metabolism | 81/121 | 0.000404805 | 0.002175826 | 0.02 | Down |
| mmu00760 | Nicotinate and nicotinamide metabolism | 11/41 | 0.000459649 | 0.002325283 | 0.04 | Up |
| mmu00860 | Porphyrin and chlorophyll metabolism | 19/43 | 0.000549279 | 0.002624333 | 0.04 | Up |
| mmu00360 | Phenylalanine metabolism | 7/23 | 0.000744591 | 0.003370253 | 0.01 | Up |
| mmu00630 | Glyoxylate and dicarboxylate metabolism | 20/32 | 0.000794408 | 0.003415956 | 0.02 | Down |
| mmu00480 | Glutathione metabolism | 28/72 | 0.001037994 | 0.004250832 | 0.02 | Down |

*Table 2 continued on next page*

Table 2 continued

| ID | Metabolic pathway | Number of detected proteins | p-Value | padj | avg.logfc | Direction |
|---|---|---|---|---|---|---|
| mmu00061 | Fatty acid biosynthesis | 12/19 | 0.00119934 | 0.004323466 | 0.04 | Up |
| mmu00052 | Galactose metabolism | 17/32 | 0.001212032 | 0.004323466 | 0.02 | Down |
| mmu00350 | Tyrosine metabolism | 12/40 | 0.001275764 | 0.004323466 | 0.02 | Down |
| mmu00900 | Terpenoid backbone biosynthesis | 8/23 | 0.001283046 | 0.004323466 | 0.01 | Down |
| mmu00140 | Steroid hormone biosynthesis | 12/92 | 0.001307094 | 0.004323466 | 0.02 | Down |
| mmu00511 | Other glycan degradation | 11/18 | 0.001620057 | 0.005160182 | 0.03 | Down |
| mmu00520 | Amino sugar and nucleotide sugar metabolism | 29/51 | 0.001922509 | 0.005904848 | 0.02 | Down |
| mmu00040 | Pentose and glucuronate interconversions | 9/35 | 0.002291756 | 0.006796243 | 0.02 | Down |
| mmu00620 | Pyruvate metabolism | 34/44 | 0.004741112 | 0.013428366 | 0.02 | Down |
| mmu00524 | Neomycin, kanamycin, and gentamicin biosynthesis | 3/5 | 0.004840458 | 0.013428366 | 0.02 | Down |
| mmu00053 | Ascorbate and aldarate metabolism | 9/31 | 0.005220665 | 0.013429037 | 0.01 | Down |
| mmu00830 | Retinol metabolism | 8/97 | 0.005279453 | 0.013429037 | 0.01 | Down |
| mmu00531 | Glycosaminoglycan degradation | 10/21 | 0.005309154 | 0.013429037 | 0.02 | Down |
| mmu00450 | Selenocompound metabolism | 9/17 | 0.006951355 | 0.017080471 | 0.02 | Down |
| mmu00250 | Alanine, aspartate, and glutamate metabolism | 17/39 | 0.007955263 | 0.019004239 | 0.02 | Down |
| mmu01230 | Biosynthesis of amino acids | 45/79 | 0.008873164 | 0.020624111 | 0.02 | Down |
| mmu01212 | Fatty acid metabolism | 40/62 | 0.009336325 | 0.021129578 | 0.03 | Down |
| mmu00500 | Starch and sucrose metabolism | 14/34 | 0.011170016 | 0.024631316 | 0.02 | Down |
| mmu00514 | Other types of O-glycan biosynthesis | 15/43 | 0.011876496 | 0.025349557 | 0.04 | Up |
| mmu01210 | 2-Oxocarboxylic acid metabolism | 11/20 | 0.01227639 | 0.025349557 | 0.01 | Down |
| mmu00650 | Butanoate metabolism | 13/28 | 0.012404185 | 0.025349557 | 0.01 | Down |
| mmu00670 | One carbon pool by folate | 9/19 | 0.012674778 | 0.025349557 | 0.05 | Down |

Table 2 continued on next page

Table 2 continued

| ID | Metabolic pathway | Number of detected proteins | p-Value | padj | avg.logfc | Direction |
|---|---|---|---|---|---|---|
| mmu00310 | Lysine degradation | 20/64 | 0.01401052 | 0.027384197 | 0.02 | Down |
| mmu00590 | Arachidonic acid metabolism | 9/86 | 0.015253177 | 0.029150517 | 0.01 | Down |
| mmu00770 | Pantothenate and CoA biosynthesis | 8/21 | 0.016632522 | 0.031095584 | 0.02 | Down |
| mmu00592 | Alpha-linolenic acid metabolism | 3/25 | 0.017819371 | 0.032605657 | 0.03 | Up |
| mmu00780 | Biotin metabolism | 3/3 | 0.018650597 | 0.033415653 | 0.01 | Down |
| mmu00640 | Propanoate metabolism | 25/31 | 0.020182799 | 0.035173564 | 0.02 | Down |
| mmu00290 | Valine, leucine, and isoleucine biosynthesis | 2/4 | 0.02053428 | 0.035173564 | 0.02 | Down |
| mmu00920 | Sulfur metabolism | 7/11 | 0.021206387 | 0.035173564 | 0.02 | Down |
| mmu00062 | Fatty acid elongation | 15/19 | 0.021267736 | 0.035173564 | 0.02 | Down |
| mmu00604 | Glycosphingolipid biosynthesis – ganglio series | 5/15 | 0.022065753 | 0.035804807 | 0.02 | Down |
| mmu00563 | Glycosylphosphatidylinositol (GPI)-anchor biosynthesis | 8/26 | 0.023553511 | 0.036186307 | 0.04 | Down |
| mmu00750 | Vitamin B6 metabolism | 3/9 | 0.023553511 | 0.036186307 | 0.03 | Up |
| mmu00220 | Arginine biosynthesis | 7/20 | 0.024245413 | 0.036186307 | 0.02 | Down |
| mmu00270 | Cysteine and methionine metabolism | 29/53 | 0.025410276 | 0.036186307 | 0.03 | Up |
| mmu00051 | Fructose and mannose metabolism | 17/36 | 0.02551991 | 0.036186307 | 0.04 | Down |
| mmu00071 | Fatty acid degradation | 30/52 | 0.025667032 | 0.036186307 | 0.02 | Down |
| mmu00330 | Arginine and proline metabolism | 23/54 | 0.025667032 | 0.036186307 | 0.02 | Down |
| mmu00561 | Glycerolipid metabolism | 23/62 | 0.025667032 | 0.036186307 | 0.03 | Down |
| mmu00010 | Glycolysis/gluconeogenesis | 40/67 | 0.050728869 | 0.068166917 | 0.02 | Down |
| mmu00030 | Pentose phosphate pathway | 17/33 | 0.050728869 | 0.068166917 | 0.03 | Down |
| mmu00410 | Beta-alanine metabolism | 19/32 | 0.050728869 | 0.068166917 | 0.01 | Down |

## Inflammation disrupts the TCA cycle in adrenocortical cells at the levels of isocitrate dehydrogenase and SDH

Inflammation downregulates the TCA cycle and oxidative phosphorylation in inflammatory activated macrophages (*Ryan and O'Neill, 2020*), however little is known about inflammation-induced metabolic changes in other cell types. We show that TCA cycle-related gene expression was downregulated in the adrenal cortex of LPS-treated mice (*Figure 2A and B*; *Table 1*). Expression of genes encoding key TCA cycle enzymes, including SDH *Sdhb* and *Sdhc*, isocitrate dehydrogenases 2 and 3 (*Idh2* and *Idh3b*), and malate dehydrogenase 1 (*Mdh1*), was reduced in the adrenal cortex of LPS-injected mice (*Figure 2B*). Proteomic GSEA confirmed the TCA cycle downregulation in steroidogenic adrenocortical cells of LPS mice (*Figure 2C*, *Table 2*). Accordingly, CD31⁻CD45⁻ adrenocortical cells from LPS-treated mice displayed reduced *Idh1*, *Idh2*, *Sdhb,* and *Sdhc* expression (*Figure 2D and E*) and LPS treatment attenuated the IDH and SDH enzymatic activities in the adrenal cortex (*Figure 2F and G*). Additionally, immunofluorescent staining showed that IDH2 and SDHB proteins are highly expressed in SF-1⁺ (steroidogenic) cells (*Figure 2H, I*). In endothelial and immune cells of the adrenal cortex of LPS-treated mice, *Idh1* and *Idh2* gene expression was reduced, *Sdhb* gene expression was increased, while expression of *Sdhc* was unaltered (*Figure 2—figure supplement 1A and B*). Collectively, these data indicate that the reduced activity of SDH in the adrenal cortex of LPS-treated mice is mainly due to its downregulated expression in steroidogenic adrenocortical cells.

In order to confirm that inflammation disrupts the TCA cycle in adrenocortical cells, we profiled the changes in metabolite levels in the adrenal glands of PBS- and LPS-treated mice using liquid chromatography-tandem mass spectrometry (LC-MS/MS). The levels of isocitrate and succinate, as well as the ratios of isocitrate/α-ketoglutarate and succinate/fumarate were increased in the adrenal glands of LPS-treated mice (*Figure 2J–O*). Furthermore, MALDI mass spectrometry imaging (MALDI-MSI) confirmed the increased levels of isocitrate and succinate in the adrenal cortex of LPS mice (*Figure 2P and Q*). These data collectively demonstrate that inflammation disrupts IDH and SDH activities and increases the levels of their substrates isocitrate and succinate in adrenocortical cells.

## Inflammation reduces oxidative phosphorylation and increases oxidative stress in the adrenal cortex

Next, we investigated how inflammation affects mitochondrial oxidative metabolism in adrenocortical cells. GSEA of the RNA-Seq and proteomic data in the adrenal cortex and CD31⁻CD45⁻ adrenocortical cells, respectively, revealed that oxidative phosphorylation was significantly enriched among the downregulated genes (*Figure 3A*) and proteins (*Figure 3B*), and expression of a large number of oxidative phosphorylation-associated genes was reduced in the adrenal cortex of LPS mice (*Figure 3C*). In accordance, ATP levels were reduced in the adrenal gland (*Figure 3D*) and the mitochondrial membrane potential of CD31⁻CD45⁻ adrenocortical cells was decreased in mice treated with LPS (*Figure 3E*). In pro-inflammatory macrophages, a TCA cycle 'break' at the level of SDH is associated with repurposing of mitochondria from oxidative phosphorylation-mediated ATP synthesis to ROS production (*Mills et al., 2016*). EGSEA pathway analysis showed that upon LPS treatment several pathways involved in the regulation of and the cellular response to oxidative stress in the adrenal cortex were enriched at mRNA (*Table 3*) and protein level (*Table 4*). This was confirmed by increased 4-hydroxynonenal (4-HNE) staining, indicating higher oxidative stress-associated damage in the adrenal cortex of LPS-treated mice (*Figure 3F*). Antioxidant defense mechanisms are particularly important in the adrenal cortex, since electron leakage through the reactions catalyzed by CYP11A1 and CYP11B1 during glucocorticoid synthesis contributes significantly to mitochondrial ROS production (*Prasad et al., 2014*). Cells neutralize ROS to maintain their cellular redox environment by using the reducing equivalents NADPH and glutathione (*Xiao and Loscalzo, 2020*). In addition, NADPH serves as a cofactor for mitochondrial steroidogenic enzymes (*Frederiks et al., 2007*). NADPH levels and glutathione metabolism-related gene expression were significantly decreased in the adrenal glands of LPS mice (*Figure 3G and H*; *Table 1*, *Table 2*). These findings collectively suggest that inflammation in the adrenal cortex is associated with increased oxidative stress, perturbed mitochondrial oxidative metabolism, reduced antioxidant capacity, and increased ROS production.

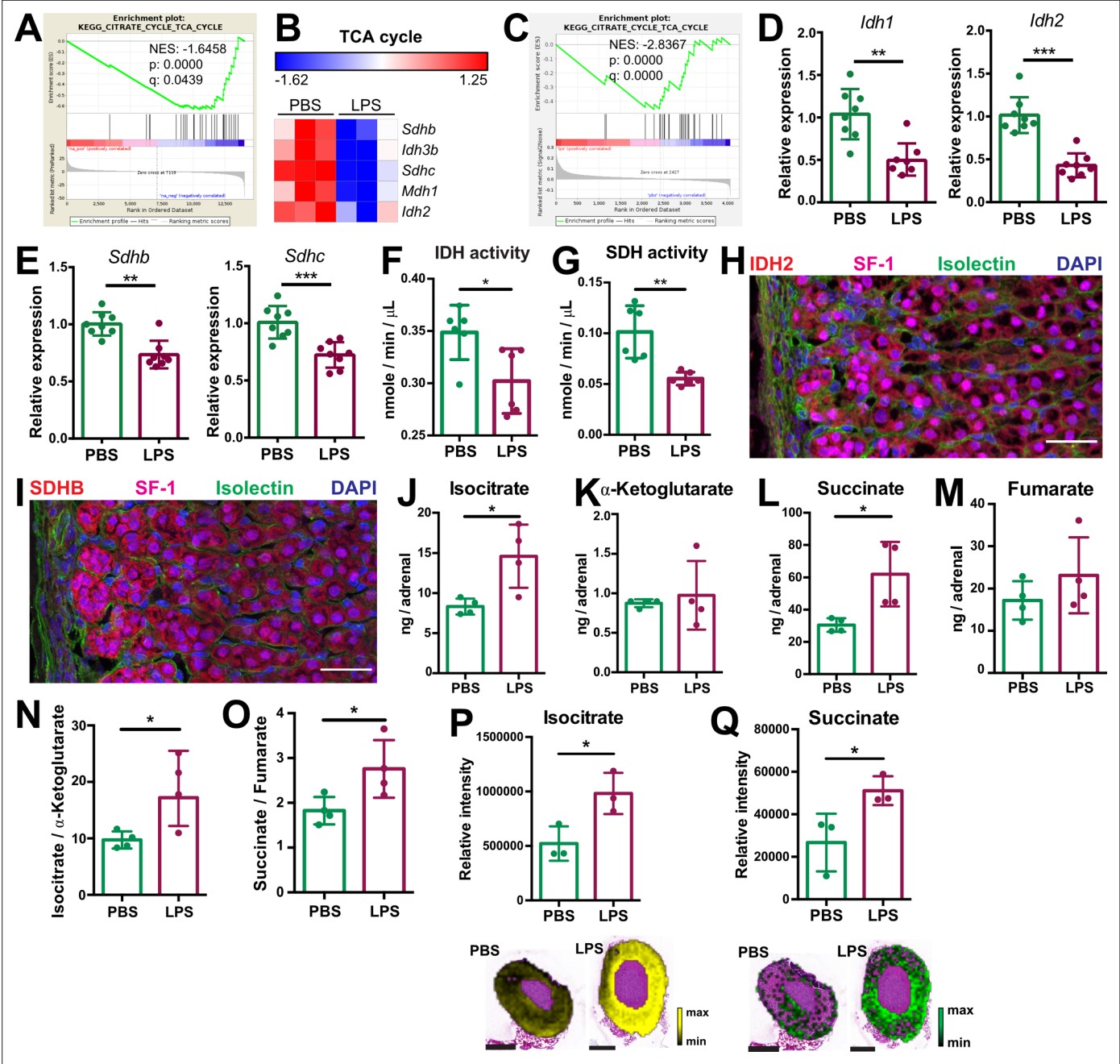

**Figure 2.** Systemic inflammation disrupts the TCA cycle in the adrenal cortex. (**A,B**) Transcriptome analysis in the microdissected adrenal gland cortex of mice treated for 6 hr with PBS or LPS (n=3 mice per group). (**A**) Gene set enrichment analysis (GSEA) for TCA cycle genes. (**B**) Heatmap of differentially expressed TCA cycle genes (padj <0.05). (**C**) GSEA analysis for TCA cycle proteins in CD31⁻CD45⁻ adrenocortical cells of mice treated for 24 hr with PBS or LPS (n=6 mice per group). (**D,E**) mRNA expression of *Idh1*, *Idh2*, *Sdhb*, and *Sdhc* in adrenocortical CD31⁻CD45⁻ cells of mice treated for 6 hr with PBS or LPS (n=8 mice per group, shown one from two experiments). (**F,G**) Quantification of IDH and SDH activities in the adrenal cortex of mice treated for 24 hr with LPS or PBS (n=6 mice per group). Values are normalized to the total protein amount in the adrenal cortex. (**H,I**) Immunofluorescence images of the adrenal gland, stained for IDH2 (red) or SDHB (red), SF-1 (magenta), Isolectin (staining endothelial cells, green), and DAPI (blue). Scale bar, 30 μm. (**J–O**) TCA cycle metabolites (isocitrate, α-ketoglutarate, succinate, fumarate) were measured by LC-MS/MS in adrenal glands of mice 24 hr after injection with PBS or LPS (n=4 mice per group, shown one from two experiments). (**P,Q**) MALDI-MSI for isocitrate and succinate in the adrenal cortex of mice treated for 24 hr with PBS or LPS (n=3 mice per group). Representative images and quantifications are shown. Scale bar, 500 μm. Data in (**D–G,J–Q**) are presented as mean ±s.d. Statistical analysis was done with two-tailed Mann-Whitney test (**D–G**) or one-tailed Mann-Whitney test (**J–Q**). *p<0.05, **p<0.01, ***p<0.001, ****p<0.0001. NES: normalized enrichment score.

The online version of this article includes the following source data and figure supplement(s) for figure 2:

*Figure 2 continued on next page*

*Figure 2 continued*

**Source data 1.** Systemic inflammation disrupts the TCA cycle in the adrenal cortex.

**Figure supplement 1.** Expression of TCA cycle genes in endothelial and immune cells of adrenal glands of LPS-treated mice.

**Figure supplement 1—source data 1.** Expression of TCA cycle genes in endothelial and immune cells of adrenal glands of LPS-treated mice.

## Increased succinate levels impair mitochondrial metabolism and steroidogenesis in adrenocortical cells

SDH is complex II of the electron transport chain (ETC), coupling succinate oxidation with the respiratory chain (*Midzak and Papadopoulos, 2016*). Inhibition of SDH function with dimethyl malonate (DMM), which is hydrolyzed to the competitive SDH inhibitor malonate (*Mills et al., 2016*; *Moosavi et al., 2020*), or treatment of adrenocortical cells with the cell-permeable succinate analog diethyl succinate (DES) increased the amount of succinate and the succinate/fumarate ratio in adrenal gland explants (*Figure 4A*) and human adrenocortical carcinoma cells NCI-H295R (*Figure 4B*). Additionally, both treatments decreased the oxygen consumption rate (OCR) and ATP production in adrenocortical cells (*Figure 4C and D*). This was associated with reduced mitochondrial membrane potential (*Figure 4E*), but not mitochondrial load (*Figure 4F*). Furthermore, DMM increased ROS (*Figure 4G*) and decreased the NADPH/NADP$^+$ ratio (*Figure 4H*), suggesting that in adrenocortical cells, as in macrophages (*Mills et al., 2016*), succinate repurposes mitochondrial metabolism from oxidative phosphorylation toward ROS production. Such changes in the mitochondrial function were not observed when inhibiting IDH activity with enasidenib (AG221) (*Yen et al., 2017*; *Figure 4I–K*). AG221 increased isocitrate and the isocitrate/α-ketoglutarate ratio (*Figure 4I*), but did not affect OCR (*Figure 4J*) or the mitochondrial membrane potential (*Figure 4K*).

Key steps of steroidogenesis take place in the mitochondria (*Midzak and Papadopoulos, 2016*), thus, we asked whether disruption of SDH activity affects steroidogenic function. We inhibited SDH activity with DMM in human and mouse adrenocortical cells, and induced glucocorticoid production by forskolin or ACTH, respectively. SDH inhibition considerably impaired glucocorticoid and progesterone production in mouse primary adrenocortical cells (*Figure 5A and B*), adrenal gland explants (*Figure 5C–E*), and human adrenocortical NCI-H295R cells (*Figure 5—figure supplement 1A and B*). Similarly, DES diminished glucocorticoid production in mouse (*Figure 5A and B*) and human adrenocortical cells (*Figure 5—figure supplement 1A and B*). Confirming these data, *Sdhb* silencing (*Figure 5—figure supplement 2A and B*) impaired glucocorticoid synthesis in mouse (*Figure 5F–H*) and human adrenocortical cells (*Figure 5—figure supplement 1C*), implying that proper adrenocortical steroidogenesis relies on intact SDH activity. Recently it was shown that SDH activity and intracellular succinate are required for CYP11A1-mediated pregnenolone synthesis, the first step of steroidogenesis (*Bose et al., 2020*). Adding to this knowledge, our data demonstrate that increasing succinate concentrations impair steroidogenesis (*Figure 5—figure supplement 1D–F*). Moreover, the proton gradient uncoupler FCCP (*Figure 5I*) and the ATP synthase inhibitor oligomycin (*Figure 5J–M*) both strongly reduced steroidogenesis in adrenocortical cells (*Figure 5I–M*), demonstrating the well-established requirement of intact mitochondrial membrane potential and ATP generation for steroidogenic function (*Bose et al., 2020*; *King et al., 1999*). We also asked whether oxidative stress mediates the effect of SDH inhibition on steroidogenesis. Reducing ROS with the antioxidant analog of vitamin E Trolox (*Figure 5N*) partially reversed the effect of DMM on cortisol and 11-deoxycortisol production (*Figure 5O–P*), suggesting that increased ROS (*Figure 4G*) contributes to impairment of steroidogenesis upon SDH blockage. In accordance, DMM and DES downregulated the expression of *Cyp11a1* and *Cyp11b1* (*Figure 5Q and R*), that catalyze the conversion of cholesterol to pregnenolone and the final step of corticosterone/cortisol production, respectively (*Midzak and Papadopoulos, 2016*; *Payne and Hales, 2004*). However, the corticosterone/11-deoxycorticosterone ratio reflecting CYP11B1 activity was not affected by *Sdhb* silencing (*Figure 5—figure supplement 1G*). Importantly, treatment of adrenal gland explants with LPS reduced corticosterone secretion in response to ACTH, similar to DMM of DES (*Figure 5S*), albeit without affecting the corticosterone/11-deoxycorticosterone ratio (*Figure 5—figure supplement 1H*). In contrast to SDH blockage, inhibition of IDH activity with AG221 (*Figure 4I*) did not alter glucocorticoid production in mouse adrenocortical cells (*Figure 5—figure supplement 3A and B*), adrenal gland explants (*Figure 5—figure supplement 3C and D*), or human adrenocortical cells (*Figure 5—figure supplement 3E and F*), nor did

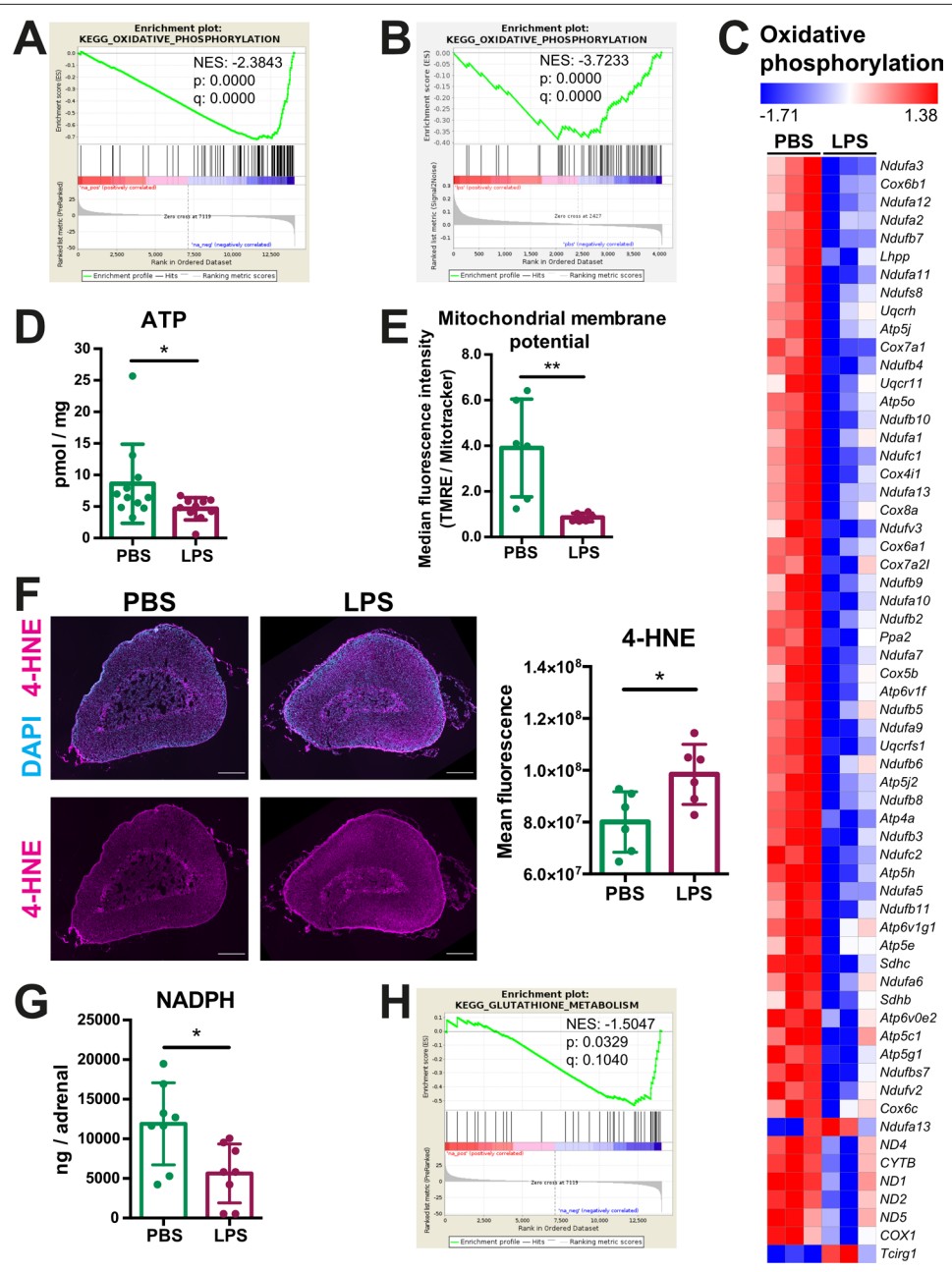

**Figure 3.** Oxidative phosphorylation is reduced and oxidative stress is increased in the adrenal cortex of LPS-treated mice. (**A**) Gene set enrichment analysis (GSEA) for oxidative phosphorylation-related genes in the adrenal cortex of mice treated for 6 hr with PBS or LPS (n=3 mice per group). (**B**) GSEA for oxidative phosphorylation-associated proteins in CD31⁻CD45⁻ adrenocortical cells of mice treated for 24 hr with PBS or LPS (n=6 mice per group). (**C**) Heatmap of differentially expressed genes related to oxidative phosphorylation (padj <0.05). (**D**) Measurement of ATP in adrenal glands of mice treated for 24 hr with PBS or LPS (n=10–11 mice per group, pooled from two experiments). (**E**) Measurement of mitochondrial membrane potential by TMRE staining and mitochondrial load by Mitotracker Green FM in CD31⁻CD45⁻adrenocortical cells of PBS or LPS mice. Data are presented as ratio of the median fluorescence intensities of TMRE to Mitotracker Green FM (n=6 mice per group). (**F**) Representative immunofluorescence images of adrenal gland sections from PBS- and LPS-treated mice (24 hr post-injection), stained for 4-hydroxynonenal (4-HNE) (magenta) and DAPI (blue). Scale bar, 300 μm. Quantification of the mean fluorescence intensity of 4-HNE staining in the adrenal cortex of PBS- or LPS-treated mice (n=6 mice per group). (**G**) NADPH measurement by liquid chromatography-tandem mass spectrometry (LC-MS/MS) in adrenal glands of mice treated with PBS or LPS for 24 hr (n=8 mice per group). Data are given as observed peak

*Figure 3 continued on next page*

*Figure 3 continued*

area intensities of NADPH. (**H**) GSEA for glutathione metabolism of RNA-Seq data in the adrenal cortex of LPS versus PBS mice (n=3 mice per group). Data in (**D–G**) present mean ± s.d. Statistical analysis was done with two-tailed Mann-Whitney test. *p<0.05, **p<0.01. NES: normalized enrichment score.

The online version of this article includes the following source data for figure 3:

**Source data 1.** Oxidative phosphorylation is reduced and oxidative stress is increased in the adrenal cortex of LPS-treated mice.

---

*Idh2* silencing in mouse adrenocortical cells (*Figure 5—figure supplement 2C*, *figure supplement 3G,H*). Taken together, these results imply that SDH but not IDH activity is required for adrenocortical steroidogenesis.

## Itaconate is not responsible for reduced SDH activity and steroidogenesis in adrenocortical cells

In inflammatory macrophages, SDH function is inhibited by itaconate (*Lampropoulou et al., 2016*), a byproduct of the TCA cycle produced from cis-aconitate in a reaction catalyzed by aconitate decarboxylase 1 (ACOD1) (*Michelucci et al., 2013*). The expression of *Acod1*, the gene encoding for ACOD1, and itaconate levels are strongly upregulated in macrophages upon inflammation (*Lampropoulou et al., 2016*). We asked whether itaconate might affect SDH activity in the adrenal cortex. *Acod1* expression was upregulated in the adrenal cortex of LPS-treated mice but this increase derived from CD45[+] cells, while *Acod1* was not expressed in CD31[-]CD45[-] adrenocortical cells (*Figure 5—figure supplement 4A*). Accordingly, LPS treatment significantly elevated itaconate levels in the CD31[+]CD45[+] fraction, while it did not increase itaconate levels in CD31[-]CD45[-] adrenocortical cells (*Figure 5—figure supplement 4B and C*). Itaconate can be secreted from LPS-stimulated

**Table 3.** ROS pathways are transcriptionally upregulated in the adrenal cortex of LPS-treated mice.
The pathway analysis of differentially expressed genes was done with the software package EGSEA and queried against the GO gene sets repository. Pathways with padj. <0.05 are shown.

| ID | Gene set | Number of expressed genes | p-Value | padj | avg.logfc | Direc tion |
|---|---|---|---|---|---|---|
| M13446 | GO Regulation of reactive oxygen species metabolic process | 271/275 | 3.75E-08 | 1.26E-06 | 1.0100 | Up |
| M13580 | GO Positive regulation of reactive oxygen species metabolic process | 182/186 | 2.33E-07 | 6.26E-06 | 1.0100 | Up |
| M16953 | GO Response to reactive oxygen species | 300/317 | 0.003422132 | 0.009035375 | 0.8600 | Up |
| M16581 | GO Cellular response to reactive oxygen species | 173/177 | 0.002537297 | 0.009035375 | 0.7600 | Up |
| M10618 | GO Negative regulation of response to reactive oxygen species | 24/24 | 0.0072384 | 0.010942115 | 0.7100 | Up |
| M15379 | GO Regulation of reactive oxygen species biosynthetic process | 145/148 | 5.90E-05 | 0.000770609 | 0.7000 | Up |
| M10827 | GO Positive regulation of reactive oxygen species biosynthetic process | 120/123 | 0.000465606 | 0.00454261 | 0.7000 | Up |
| M15990 | GO Reactive oxygen species metabolic process | 163/167 | 0.008936465 | 0.012568942 | 0.6700 | Up |
| M16764 | GO Regulation of response to reactive oxygen species | 43/43 | 0.006753207 | 0.010498628 | 0.6200 | Down |
| M16007 | GO Negative regulation of reactive oxygen species biosynthetic process | 23/23 | 0.016434387 | 0.020274996 | 0.6000 | Up |
| M12185 | GO Reactive oxygen species biosynthetic process | 32/33 | 0.006483259 | 0.01024232 | 0.5700 | Down |
| M10894 | GO Negative regulation of reactive oxygen species metabolic process | 59/59 | 0.006538654 | 0.010287661 | 0.5500 | Up |

**Table 4.** ROS-related protein expression is upregulated in the adrenal cortex of LPS-treated mice. The pathway analysis of differentially expressed proteins was done with the software package EGSEA and queried against the GO gene sets repository. Pathways with padj. <0.05 are shown.

| ID | Protein set | Number of detected proteins | p-Value | padj | avg.logfc | Direc-tion |
|---|---|---|---|---|---|---|
| M13446 | GO REGULATION OF REACTIVE OXYGEN SPECIES METABOLIC PROCESS | 62/275 | 6.30E-06 | 0.00012401 | 0.03 | Up |
| M15379 | GO REGULATION OF REACTIVE OXYGEN SPECIES BIOSYNTHETIC PROCESS | 34/148 | 9.43E-06 | 0.00014372 | 0.03 | Up |
| M13580 | GO POSITIVE REGULATION OF REACTIVE OXYGEN SPECIES METABOLIC PROCESS | 29/186 | 1.11E-05 | 0.00015778 | 0.03 | Up |
| M10827 | GO POSITIVE REGULATION OF REACTIVE OXYGEN SPECIES BIOSYNTHETIC PROCESS | 20/123 | 1.17E-05 | 0.00016203 | 0.03 | Up |
| M10894 | GO NEGATIVE REGULATION OF REACTIVE OXYGEN SPECIES METABOLIC PROCESS | 23/59 | 5.29E-05 | 0.00037856 | 0.05 | Down |
| M16007 | GO NEGATIVE REGULATION OF REACTIVE OXYGEN SPECIES BIOSYNTHETIC PROCESS | 12/23 | 0.00014126 | 0.00073167 | 0.07 | Up |
| M16581 | GO CELLULAR RESPONSE TO REACTIVE OXYGEN SPECIES | 50/177 | 0.00015378 | 0.00077551 | 0.03 | Down |
| M15990 | GO REACTIVE OXYGEN SPECIES METABOLIC PROCESS | 35/167 | 0.00037436 | 0.00150479 | 0.02 | Down |
| M16953 | GO RESPONSE TO REACTIVE OXYGEN SPECIES | 83/317 | 0.00076076 | 0.00262274 | 0.03 | Up |
| M12185 | GO REACTIVE OXYGEN SPECIES BIOSYNTHETIC PROCESS | 9/33 | 0.00192844 | 0.00545214 | 0.02 | Down |

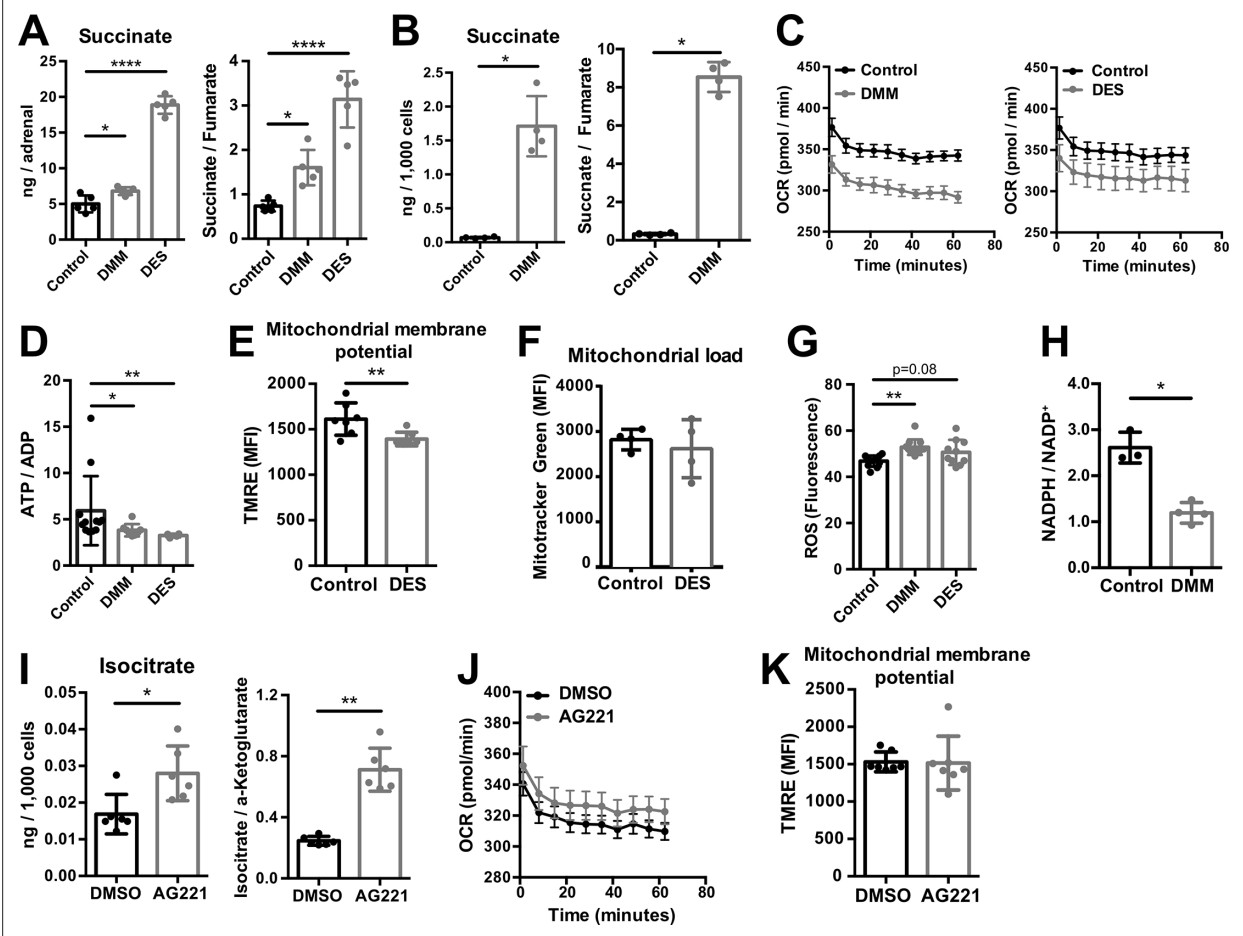

**Figure 4.** Increased succinate levels impair mitochondrial function in adrenocortical cells. (**A,B**) Succinate and fumarate levels were measured by liquid chromatography-tandem mass spectrometry (LC-MS/MS) in adrenal gland explants (**A**) and NCI-H295R cells (**B**) treated with dimethyl malonate (DMM) or diethyl succinate (DES) for 24 hr (n=5 for (**A**) and n=4 for (**B**)). (**C**) Oxygen consumption rate (OCR) measurement with Seahorse technology in NCI-H295R cells treated with DMM or DES for 24 hr (n=6). (**D**) Measurement of ATP/ADP ratio in NCI-H295R cells treated with DMM or DES for 24 hr (n=4–12). (**E,F**) TMRE and Mitotracker Green FM staining assessed by flow cytometry in NCI-H295R cells treated with DES for 4 hr, MFI is shown (n=7 for (**E**) and n=4, one from two experiments for (**F**)). (**G**) ROS measurement in NCI-H295R cells treated with DMM or DES for 2 hr (n=10–12). (**H**) Measurement of NADPH/NADP$^+$ ratio in NCI-H295R cells treated with DMM for 24 hr (n=3–4). (**I**) Isocitrate levels measured by LC-MS/MS in NCI-H295R cells treated for 24 hr with AG221 or DMSO (n=6). (**J**) OCR measurement in NCI-H295R cells treated for 24 hr with AG221 or DMSO (n=10). (**K**) TMRE staining and flow cytometry in NCI-H295R cells treated for 4 hr with AG221 or DMSO, MFI is shown (n=7). Data in (**A–B,D–I,K**) are presented as mean ± s.d. Data in (**C,J**) are presented as mean ± s.e.m. Statistical analysis was done with one-way ANOVA (**A, G**) or two-tailed (B,D,E,F,I,K) or one-tailed (**H**) Mann-Whitney test. *p<0.05, **p<0.01, ***p<0.001, ****p<0.0001.

The online version of this article includes the following source data for figure 4:

**Source data 1.** Increased succinate levels impair mitochondrial function in adrenocortical cells.

macrophages (**Lampropoulou et al., 2016**), and could thereby affect SDH activity in adrenocortical cells. Therefore, we tested whether exogenously given itaconate may affect steroidogenesis by treating primary adrenocortical cells with the cell-permeable itaconate derivative 4-octyl itaconate (4-OI). Adrenocortical cells internalized the added itaconate derivative (**Figure 5—figure supplement 4D**), which however did not alter succinate or fumarate levels or the succinate/fumarate ratio (**Figure 5—figure supplement 4E–G**), nor did it affect glucocorticoid production (**Figure 5—figure supplement 4H–I**). Additionally, SDH activity in the adrenal cortex of *Acod1*-KO mice injected with LPS was not different from that in their wild-type counterparts (**Figure 5—figure supplement 4J**). Hence, neither is itaconate produced nor does it affect SDH activity through paracrine routes in adrenocortical cells.

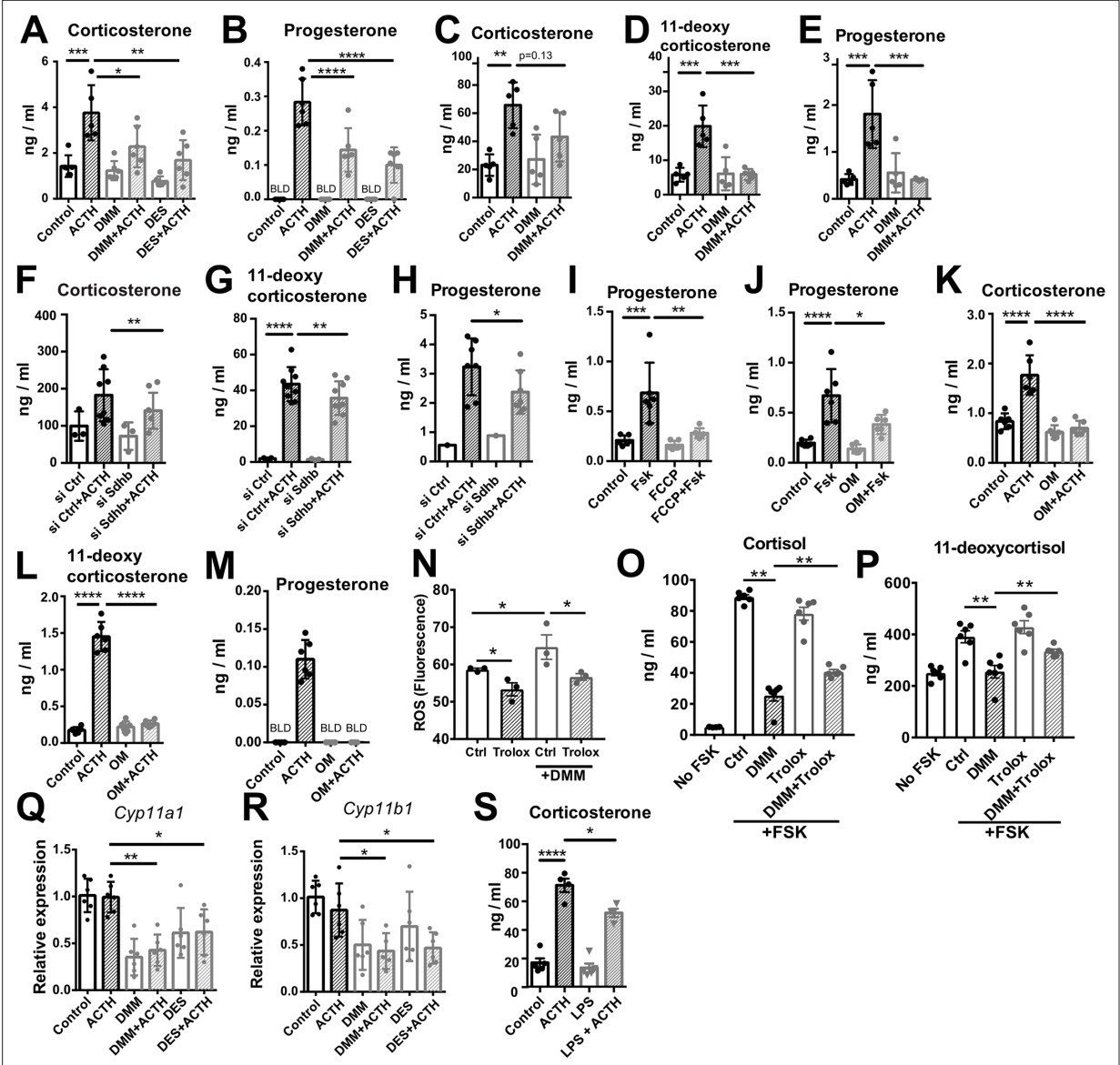

**Figure 5.** Disruption of SDH function impairs glucocorticoid production. (**A–E**) Primary adrenocortical cells (**A,B**) and adrenal explants (**C–E**) were treated for 24 hr with dimethyl malonate (DMM) or diethyl succinate (DES) and for another 45 min with adrenocorticotropic hormone (ACTH) (10 ng/ml or 100 ng/ml, respectively) (n=5–6). (**F–H**) Primary adrenocortical cells were transfected with si*Sdhb* or non-targeting siRNA (siCtrl) and 24 hr post-transfection they were treated for 45 min with ACTH (n=7–8). (**I,J**) NCI-H295R cells were treated for 24 hr with FCCP (**I**) or oligomycin (OM) (**J**) and for another 30 min with Forskolin (Fsk) (n=6). (**K–M**) Primary adrenocortical cells were treated for 24 hr with oligomycin (OM) and for another 45 min with ACTH (n=6). (**N**) ROS measurement in NCI-H295R cells pre-treated for 15 min with Trolox or control solution (DMSO) and then treated for 2 hr with DMM (n=3). (**O,P**) NCI-H295R cells pre-treated for 15 min with Trolox or DMSO were treated or not for 24 hr with DMM and Forskolin (n=6). (**Q,R**) *Cyp11a1* and *Cyp11b1* expression in primary adrenocortical cells treated for 24 hr with DMM or DES and for 45 min with ACTH (n=5–6). (**S**) Adrenal gland explants were treated for 24 hr with LPS and for 45 min with ACTH (n=4–5). Measurements of steroid hormones in (**A–M,O,P,S**) were performed in supernatants of primary adrenocortical cell cultures or adrenal gland explants by liquid chromatography-tandem mass spectrometry (LC-MS/MS). Data are presented as mean ± s.d. Statistical analysis was done with one-way ANOVA (**A–E, I–M,S**), Wilcoxon (**F,G,H**), one-tailed Mann-Whitney (**N**), or two-tailed Mann-Whitney test (**O–R**). *p<0.05, **p<0.01, ***p<0.001, ****p<0.0001. BLD = below level of detection.

The online version of this article includes the following source data and figure supplement(s) for figure 5:

**Source data 1.** Disruption of SDH function impairs glucocorticoid production.

**Figure supplement 1.** High succinate levels impair glucocorticoid production in adrenocortical cells.

**Figure supplement 1—source data 1.** High succinate levels impair glucocorticoid production in adrenocortical cells.

**Figure supplement 2.** SiRNA silencing efficiencies.

*Figure 5 continued on next page*

*Figure 5 continued*

**Figure supplement 2—source data 1.** SiRNA silencing efficiencies.

**Figure supplement 3.** Disruption of IDH function does not affect glucocorticoid production.

**Figure supplement 3—source data 1.** Disruption of IDH function does not affect glucocorticoid production.

**Figure supplement 4.** Itaconate does not affect SDH activity or steroidogenesis in adrenocortical cells.

**Figure supplement 4—source data 1.** Itaconate does not affect SDH activity or steroidogenesis in adrenocortical cells.

## IL-1β downregulates SDHB expression and steroidogenesis in a DNMT1-dependent manner

Systemic inflammation induces substantial leukocyte recruitment in the adrenal gland, accompanied by elevated production of pro-inflammatory cytokines (*Chen et al., 2019a*; *Kanczkowski et al., 2013b*). Among them, IL-1β is highly produced by inflammatory monocytes and macrophages (*Netea et al., 2010*). RNA-Seq in the adrenal cortex, including recruited immune cells, showed increased expression of *Il1b* in LPS- compared to PBS-injected mice (log2fold change [fc] = 1.46, padj = 0.019). Furthermore, there was significant positive enrichment of genes associated with IL-1β secretion in the adrenal cortex of mice treated with LPS (*Figure 6A*). The IL-1β receptor *Il1r1* is expressed in CD31⁻CD45⁻ adrenocortical cells and its expression was upregulated in adrenocortical cells sorted from LPS-treated mice (*Figure 6B*). In accordance, proteins related to IL-1β signaling were positively enriched in CD31⁻CD45⁻ adrenocortical cells of LPS mice (*Figure 6C*). Essentially, IL-1β, but not IL-6 or TNFα, reduced *SDHB* expression in NCI-H295R cells (*Figure 6D*). Moreover, IL-1β decreased the ATP/ADP ratio (*Figure 6E*) and impaired ACTH-induced steroidogenesis in adrenocortical cells (*Figure 6F–H*), mimicking the effects of LPS (*Figures 3D and 5S*) and DMM/DES (*Figures 4D and 5A–E*).

One way of transcriptional gene repression is covalent attachment of methyl groups on the cytosine 5′ position within the gene promoter sequence, a reaction catalyzed by DNA methyltransferases (*Suzuki and Bird, 2008*). Proteomics revealed significant upregulation of DNMT1 in CD31⁻CD45⁻ adrenocortical cells of LPS mice (log2fc = 0.421, padj = 0.015), which we confirmed by western blot analysis (*Figure 6I*). IL-1β increased DNA methylation of the *SDHB* promoter (*Figure 6J*) and the effect of IL-1β was blunted by *DNMT1* silencing (*Figure 6K*, *Figure 5—figure supplement 2D*). In accordance, *Dnmt1* repression restored *Sdhb* expression (*Figure 6L*) and reduced the succinate/fumarate ratio in IL-1β-treated adrenocortical cells (*Figure 6M*). Moreover, IL-1β decreased OCR in a DNMT1-dependent manner (*Figure 4N*). Accordingly, the inhibitory effect of IL-1β on steroidogenesis was restored by *Dnmt1* silencing (*Figure 6O–Q*).

Lastly, we set out to validate the impact of IL-1β on adrenal gland function in vivo. To this end, LPS-challenged mice were treated with Raleukin, an IL-1R antagonist, or control solution. Raleukin increased SDH activity in the adrenal cortex (*Figure 6R*), reduced succinate levels and the succinate/fumarate ratio in the adrenal gland (*Figure 6S, T*), and increased corticosterone plasma levels in LPS-treated mice (*Figure 6U*), thereby validating the hypothesis that IL-1β negatively regulates SDH function and steroidogenesis in the inflamed adrenal cortex.

## Discussion

Glucocorticoid production in response to inflammation is essential for survival. The adrenal gland shows great resilience to damage induced by inflammation due to its strong regenerative capacity (*Kanczkowski et al., 2013b*; *Lyraki and Schedl, 2021*; *Mateska et al., 2020*). This maintains glucocorticoid release during infection or sterile inflammation, which is vital to restrain and resolve inflammation (*Alexaki and Henneicke, 2021b*; *Chrousos, 1995*). However, severe sepsis is associated with adrenocortical impairment (*Annane et al., 2006*; *Annane et al., 2000*; *Boonen et al., 2015*; *Boonen et al., 2014*; *den Brinker et al., 2005*; *Jennewein et al., 2016*). Here, we used an LPS mouse model to study the extent to which cell metabolic changes in the inflamed adrenal cortex affect adrenocortical function. Due to its reproducibility, LPS-induced systemic inflammation is a widely used model, which however comes with certain limitations. Being a component of gram-negative bacteria, LPS does not trigger immune reactions similar to these caused by gram-positive microorganisms or in polymicrobial sepsis. LPS is a single pathogen-associated molecular pattern (PAMP) which specifically triggers toll-like receptor 4, while sepsis is driven by a wide range of PAMPs. Moreover, LPS-induced

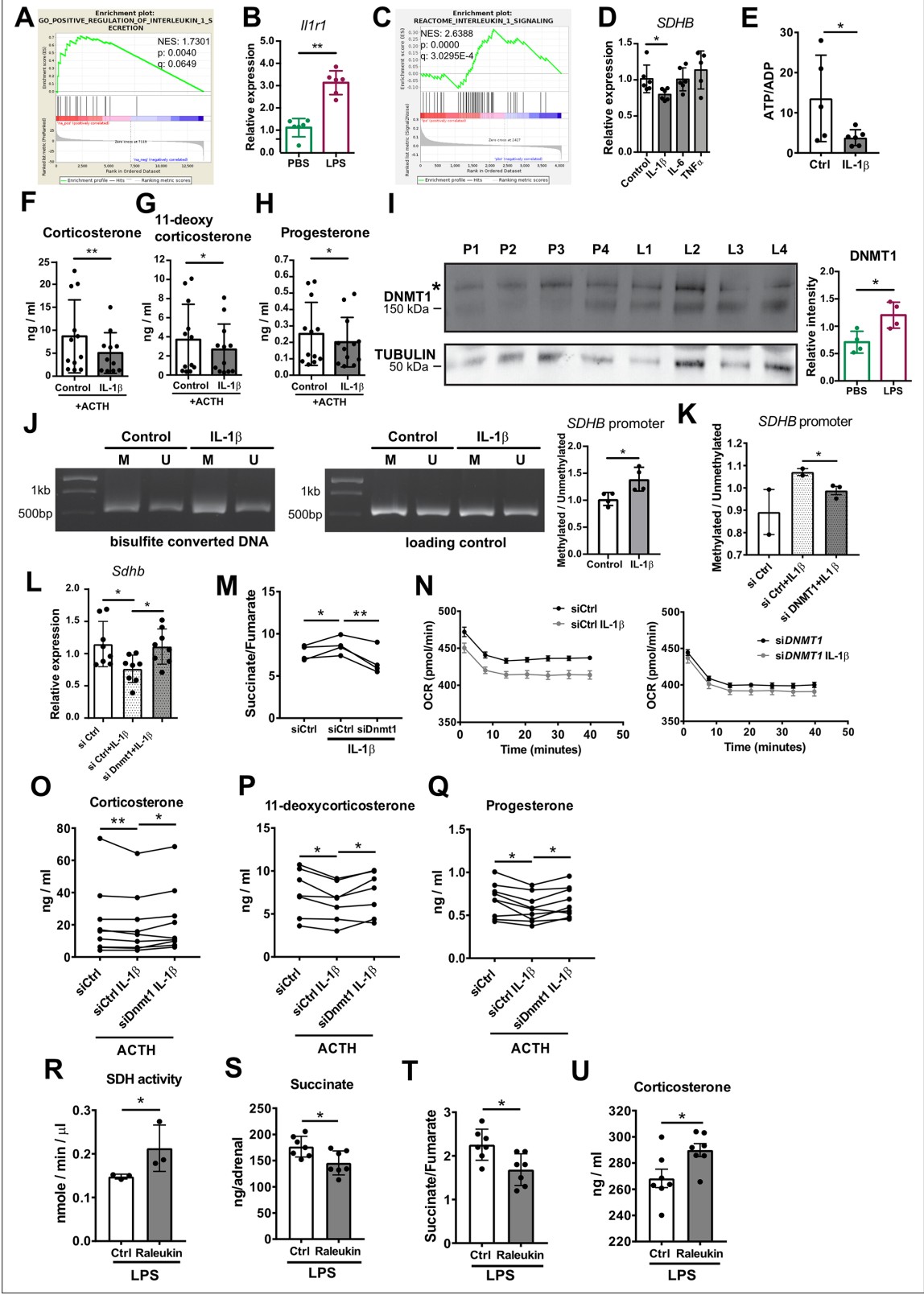

**Figure 6.** IL-1β reduces SDHB expression and adrenocortical steroidogenesis in a DNA methyltransferase 1 (DNMT1)-dependent manner. (**A**) Gene set enrichment analysis (GSEA) for genes related to positive regulation of IL-1β secretion in the adrenal cortex of mice treated for 6 hr with PBS or LPS (n=3 mice per group). (**B**) *Il1r1* expression in CD31⁻CD45⁻ adrenocortical cells of mice 6 hr post-injection with PBS or LPS (n=6 mice per group). (**C**) GSEA for proteins related to IL-1β signaling in CD31⁻CD45⁻ adrenocortical cells of mice treated for 24 hr with PBS or LPS (n=6 mice per group). (**D**) *SDHB*

*Figure 6 continued on next page*

*Figure 6 continued*

expression in NCI-H295R cells treated for 2 hr with IL-1β, IL-6, or TNFα (n=5–6). (**E**) Measurement of ATP/ADP ratio in NCI-H295R cells treated for 24 hr with IL-1β (n=5–6). (**F–H**) Primary adrenocortical cells were treated for 6 hr with IL-1β and for another 45 min with adrenocorticotropic hormone (ACTH) (10 ng/ml) (n=11–12). Steroid hormones were measured in the culture supernatant by liquid chromatography-tandem mass spectrometry (LC-MS/MS). (**I**) Western blot analysis for DNMT1 in CD31⁻CD45⁻ adrenocortical cells 24 hr after injection of PBS (**P**) or LPS (**L**) (n=4 mice per group), α-TUBULIN was used as loading control. The asterisk (*) depicts an unspecific band. Quantification of the western blot is shown as relative intensity of DNMT1 to α-TUBULIN. (**J**) NCI-H295R cells were treated for 2 hr with IL-1β; representative gel electrophoresis images of bisulfite converted and non-treated DNA (M – methylated, U – unmethylated) are shown. The ratio of methylated to unmethylated *SDHB* promoter was assayed after bisulfite conversion (n=4). (**K**) NCI-H295R cells were transfected with si*DNMT1* or siCtrl and 24 hr post-transfection they were treated for 2 hr with IL-1β. The ratio of methylated to unmethylated *SDHB* promoter was quantified (n=2–3). (**L**) *Sdhb* expression in primary adrenocortical cells transfected with si*Dnmt1* or siCtrl and 24 hr post-transfection treated for 6 hr with IL-1β (n=8). (**M**) Primary adrenocortical cells were transfected with si*Dnmt1* or siCtrl and 6 hr post-transfection they were treated for 18 hr with IL-1β (n=4). Succinate and fumarate were measured by LC-MS/MS. (**N**) Oxygen consumption rate (OCR) measurement in NCI-H295R cells transfected with si*DNMT1* or siCtrl and 24 hr post-transfection treated for 24 hr with IL-1β (n=8). (**O–Q**) Primary adrenocortical cells were transfected with si*Dnmt1* or siCtrl, 6 hr post-transfection they were treated for 18 hr with IL-1β and subsequently they were stimulated for 45 min with ACTH (n=7–9). Steroid hormones were measured in the cell culture supernatant by LC-MS/MS. (**R**) Mice were simultaneously injected with Raleukin or control solution and LPS and 24 hr later SDH activity was measured in isolated adrenal cortices (n=3 mice per group). (**S–T**) Mice were treated with Raleukin or control solution together with LPS and 24 hr post-injection succinate and fumarate levels were determined in the adrenal glands (n=7 mice per group). (**U**) Mice were treated with Raleukin or control solution together with LPS and 6 hr later corticosterone plasma levels were determined by LC-MS/MS (n=7 mice per group). Data in (**B,D–L,R–U**) are presented as mean ± s.d. Statistical analysis was done with Mann-Whitney (**B,D,I,J,N,R–U**), unpaired t-test (**E,K**), paired t-test, (**M**) and Wilcoxon test (**F–H,L,O–Q**). *p<0.05, **p<0.01. NES: normalized enrichment score. Full unedited blots and gels are available in *Figure 6—source data 1* (**I,J**).

The online version of this article includes the following source data for figure 6:

**Source data 1.** IL-1β reduces SDHB expression and adrenocortical steroidogenesis in a DNA methyltransferase 1 (DNMT1)-dependent manner.

systemic inflammation causes a rapid increase in cytokine levels followed by fast resolution of inflammation, while clinical sepsis is characterized by prolonged elevation of cytokine levels (*Lewis et al., 2016*). Despite its limitations, its high reproducibility compared to other models, such as the cecal slurry model, makes it suitable for mechanistic studies, such as the present.

Here, we show that the inflamed adrenal cortex undergoes cellular metabolic reprograming which involves perturbations in the TCA cycle and oxidative phosphorylation, leading to impaired steroidogenesis. Our findings provide a mechanistic explanation of inflammation-related impaired adrenocortical steroidogenesis through cell metabolic reprogramming of steroidogenic adrenocortical cells. Specifically, we demonstrate that IL-1β reduces *SDHB* expression through DNMT1-dependent DNA methylation of the *SDHB* promoter. Several studies have shown that inflammation promotes DNA methylation and thereby regulates gene expression (*Koos et al., 2020*; *Li et al., 2020*; *Morante-Palacios et al., 2021*; *Rodriguez et al., 2019*; *Weiss et al., 2021*). Particularly IL-1β was demonstrated to increase DNA methylation in different genes in a cell type-specific manner (*Li et al., 2020*; *Seutter et al., 2020*). In accordance, DNMT1 expression was shown to increase upon acute inflammation in human peripheral blood mononuclear cells or mouse spleens (*Cao et al., 2020*; *Koos et al., 2020*), as well as in fibroblasts treated with IL-1β (*Seutter et al., 2020*). Moreover, reduced *SDH* promoter methylation associates with enhanced SDHB expression and reduced succinate levels in villi from individuals with recurrent spontaneous abortion (*Wang et al., 2021*). These reports stand in accordance with our findings showing regulation of SDHB expression through its promoter methylation by an IL-1β-DNMT1 axis in steroidogenic adrenocortical cells. In contrast, itaconate, which was shown to reduce SDH activity in macrophages (*Lampropoulou et al., 2016*), does not regulate SDH in adrenocortical cells.

Accumulation of succinate leads to impaired oxidative phosphorylation and ATP synthesis, coupled to reduced steroidogenesis. Intact mitochondrial membrane potential and ATP generation are essential requirements for steroidogenic function (*Bose et al., 2020*; *King et al., 1999*). We confirmed this by treatment of adrenocortical cells with the mitochondrial uncoupler FCCP and the ATP synthase inhibitor oligomycin, both of which diminished steroidogenesis. Interestingly, a switch from the canonical toward a non-canonical TCA cycle, involving the metabolism of mitochondrially derived citrate to acetyl-CoA, was recently described and may be activated in inflammation (*Arnold et al., 2022*; *Mateska and Alexaki, 2022*). It remains to be elucidated whether a shift to the non-canonical TCA cycle might regulate steroidogenesis.

Intact SDH function was recently shown to be required for activation of the first steroidogenic enzyme, cytochrome P450-side-chain-cleavage (SCC, CYP11A1), which converts cholesterol to pregnenolone (*Bose et al., 2020*; *King et al., 1999*). Accordingly, we show that production of progesterone, the direct derivative of pregnenolone, is diminished upon SDH inhibition. These data suggest that impairment of SDH function may disrupt these first steps of steroidogenesis, thereby diminishing production of all downstream adrenocortical steroids.

SDH regulates ETC-mediated ROS formation: SDH inhibition or increased succinate levels augment ROS generation in tumors and macrophages (*Guzy et al., 2008*; *Hadrava Vanova et al., 2020*; *Mills et al., 2016*; *Ralph et al., 2011*; *Selak et al., 2005*). Similarly, we show that SDH inhibition or high succinate levels in adrenocortical cells lead to increased ROS levels at the expense of mitochondrial oxidative function and ATP production, while ROS scavenging partially restores steroidogenesis. Adrenocortical disorders such as triple A syndrome and familial glucocorticoid deficiency can be driven by increased oxidative stress in the adrenal cortex (*Prasad et al., 2014*). In fact, mutations in

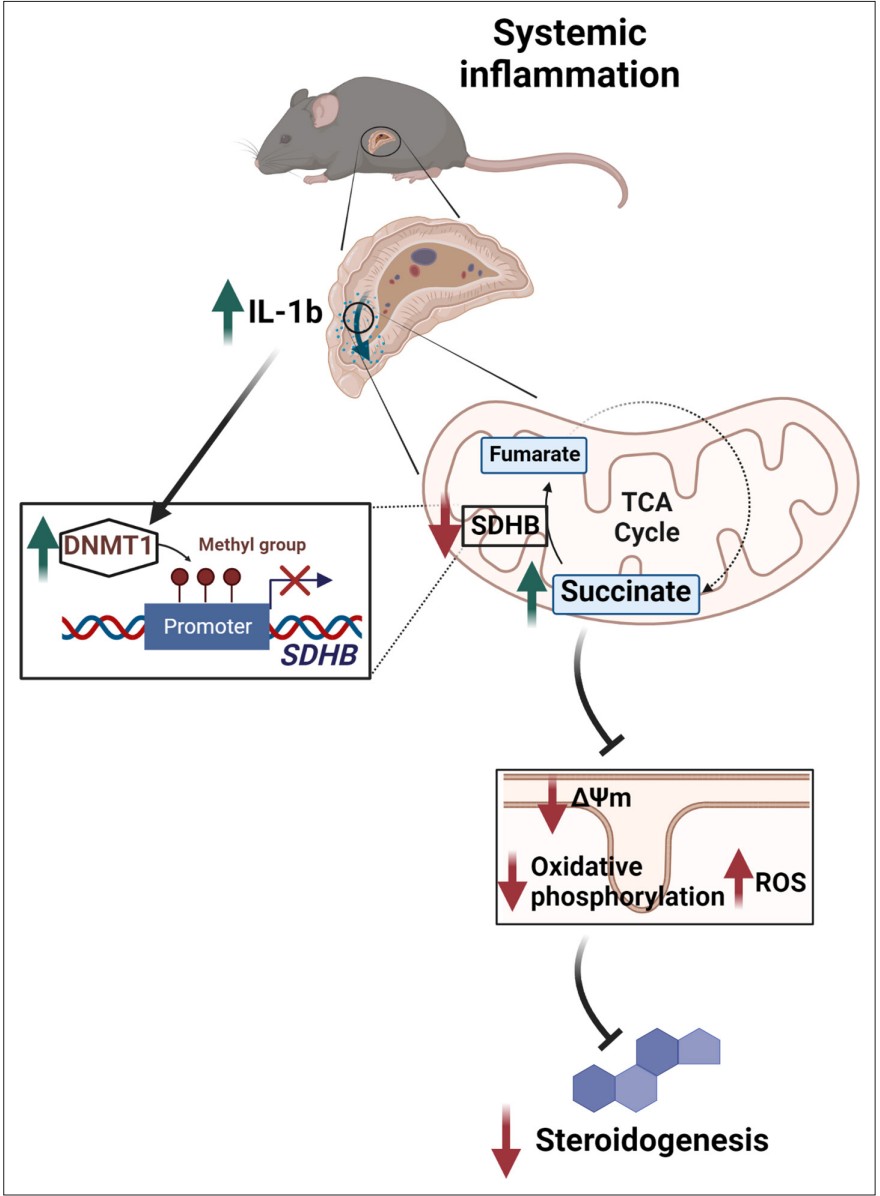

**Figure 7.** Illustration of the regulation of adrenocortical steroidogenesis by inflammation. IL-1β reduces SDHB expression through upregulation of DNA methyltransferase 1 (DNMT1) and methylation of the SDHB promoter. Consequently, increased succinate levels impair oxidative phosphorylation and increase ROS production, leading to reduced steroidogenesis.

genes encoding for proteins conferring antioxidant protection were implicated in the development of adrenocortical deficiencies (**Prasad et al., 2014**). Hence, SDH dysfunction leading to oxidative stress may be an important component of the pathophysiology of adrenocortical insufficiency, a notion which merits further investigation.

In conclusion, we demonstrate that tight regulation of succinate levels is essential for normal steroidogenesis, while disruption of SDH expression through the IL-1β-DNMT1 axis contributes to adrenocortical dysfunction (**Figure 7**). This study expands the current knowledge on the regulation of glucocorticoid production and identifies potential targets for therapeutic interventions.

# Materials and methods

**Key resources table**

| Reagent type (species) or resource | Designation | Source or reference | Identifiers | Additional information |
|---|---|---|---|---|
| Gene (*Mus musculus*) | C57BL/6J | The Jackson Laboratory | Stock#000664 RRID:MGI:3028467 | |
| Gene (*Mus musculus*) | C57BL/6NJ-Acod1<sup>em1(IMPC)J</sup>/J | The Jackson Laboratory | Strain #:029340 RRID:IMSR_JAX:029340 | |
| Cell line (*Homo sapiens*) | NCI-H295R | ATCC | CRL-2128 | |
| Chemical compound, drug | Ultrapure LPS, *E. coli* 0111:B4 | InVivoGen | tlrl-3pelps | For in vivo |
| Chemical compound, drug | Raleukin | MedChemExpress | Art. -Nr.: HY-108841 | |
| Chemical compound, drug | Ultrapure lipopolysaccharide from *E. coli* K12 | InVivoGen | tlrl-peklps | For in vitro |
| Chemical compound, drug | DMM | Sigma-Aldrich | 136441 | |
| Chemical compound, drug | DES | Sigma-Aldrich | 112402 | |
| Chemical compound, drug | FCCP | Agilent Technologies | Seahorse XFp Cell Mito Stress Test Kit 103010-100 | |
| Chemical Compound, drug | Oligomycin | Agilent Technologies | Seahorse XFp Cell Mito Stress Test Kit 103010-100 | |
| Chemical compound, drug | Enasidenib (AG-221) | Selleckchem | S8205 | |
| Chemical compound, drug | 4-Octyl-itaconate | Cayman Chemical | 25374 | |
| Chemical compound, drug | Trolox | Abcam | ab120747 | |
| Peptide, recombinant protein (human) | IL-1β | PeproTech | 200-01B | |
| Peptide, recombinant protein (mouse) | IL-1β | PeproTech | 211-11B | |
| Peptide, recombinant protein (human) | IL-6 | PeproTech | 200-06 | |
| Peptide, recombinant protein (human) | TNFα | PeproTech | 300-01A | |
| Peptide, recombinant protein (mouse) | ACTH | Sigma-Aldrich | A0298 | |

*Continued on next page*

*Continued*

| Reagent type (species) or resource | Designation | Source or reference | Identifiers | Additional information |
|---|---|---|---|---|
| Chemical compound, drug | Forskolin | Sigma-Aldrich | F3917 | |
| Transfected construct (human) | siRNA to *SDHB* (ON-TARGETplus siRNA SMARTpool) | Dharmacon/Thermo Fisher Scientific | L-011773-02-0005 | |
| Transfected construct (human) | siRNA to *DNMT1* (ON-TARGETplus siRNA SMARTpool) | Dharmacon/Thermo Fisher Scientific | L-004605-00-0005 | |
| Transfected construct (mouse) | siRNA to *Sdhb* (ON-TARGETplus siRNA SMARTpool) | Dharmacon/Thermo Fisher Scientific | L-042339-01-0005 | |
| Transfected construct (mouse) | siRNA to *Dnmt1* (ON-TARGETplus siRNA SMARTpool) | Dharmacon/Thermo Fisher Scientific | L-056796-01-0005 | |
| Sequence-based reagent | See *Table 5* | This paper | qPCR primers | See *Table 5* |
| Antibody | Anti-SDHB (Rabbit polyclonal) | Sigma-Aldrich | HPA002868 | 1:1000 for WB 1:300 for IF |
| Antibody | anti-IDH2 (Rabbit polyclonal) | Sigma-Aldrich | HPA007831 | 1:50 for IF |
| Antibody | Anti-DNMT1 (Rabbit monoclonal) | Cell Signaling | #5032 | 1:1000 |
| Antibody | Anti-Tubulin (Mouse monoclonal) | Sigma-Aldrich | T5186 | 1:3000 |
| Antibody | Anti-β-Actin (Rabbit polyclonal) | Cell Signaling | #4967 | 1:1000 |
| Antibody | Anti-SF-1 (Mouse monoclonal) | TransGenic Inc | KO610 | 1:100 |
| Commercial assay or kit | ATP measurement | Abcam | ab83355 | |
| Commercial assay or kit | ATP/ADP measurement | Sigma-Aldrich | MAK135 | |
| Commercial assay or kit | DCFDA/H2DCFDA Cellular ROS Detection Assay Kit | Abcam | ab113851 | |
| Commercial assay or kit | NADP/NADPH Assay | Abcam | ab176724 | |
| Commercial assay or kit | SDH activity | Sigma-Aldrich | MAK197 | |
| Commercial assay or kit | IDH activity | Abcam | ab102528 | |
| Commercial assay or kit | Seahorse XFp Cell Mito Stress Test Kit | Agilent Technologies | 103010-100 | |
| Commercial assay or kit | EZ DNA Methylation Kit | Zymo Research | D5001 | |
| Software, algorithm | ImageJ software | ImageJ (http://imagej.nih.gov/ij/) | RRID:SCR_003070 | |
| Software, algorithm | GraphPad Prism 7.04 software | GraphPad Prism (https://graphpad.com) | RRID:SCR_015807 | |
| Software, algorithm | Morpheus | Broad Institute | https://software.broadinstitute.org/morpheus/ | |
| Software, algorithm | STAR Aligner | *Dobin et al., 2013* | | |
| Software, algorithm | Mouse Genome version GRCm38 (release M12 GENCODE) | *Anders et al., 2015* | | |
| Software, algorithm | DESeq2_1.8.1 | *Anders and Huber, 2010* | | |

*Continued on next page*

*Continued*

| Reagent type (species) or resource | Designation | Source or reference | Identifiers | Additional information |
|---|---|---|---|---|
| Software, algorithm | ggplot2_1.0.1 | *Wickham, 2009* | | |
| Software, algorithm | GSEA | *Subramanian et al., 2005* | | |
| Software, algorithm | EGSEA | *Alhamdoosh et al., 2017* | | |
| Software, algorithm | Mass Spectrometry Downstream Analysis Pipeline (MS-DAP) (version beta 0.2.5.1) (https://github.com/ftwkoopmans/msdap) | *Hondius et al., 2021* | | |
| Software, algorithm | R/Bioconductor, 'impute' command running of 'DEP' | *Zhang et al., 2018* | | |
| Other | TMRE | Thermo Fisher | T669 | 2.5 µM for dissociated adrenocortical cells, 100 nM for NCI-H295R cells |
| Other | Mitotracker Green | Thermo Fisher | M7514 | 0.25 µM for dissociated adrenocortical cells, 100 nM for NCI-H295R cells |
| Other | DAPI stain | Roche, Sigma-Aldrich | 10236276001 | 1:10,000 |
| Other | Lectin Esculentum DyLight488 | Vector Laboratories | DL-1174 | 1:300 |
| Other | 4-Hydroxynonenal | Abcam | ab48506 | 1:200 |

## Animal experiments

Eight- to twelve-week-old male C57BL/6J mice (purchased from Charles River) were injected i.p. with 1 mg/kg LPS (LPS-EB Ultrapure; InVivoGen) or PBS, and sacrificed after 6 hr (for gene expression analyses) or 24 hr (for all other analyses). In some experiments, mice were simultaneously i.p. injected with Raleukin (Anakinra, 10 mg/kg, MedChemExpress) or with same amount of control solution and LPS. *Acod1*[-/-] and littermate control mice were injected with 3 mg/kg LPS and sacrificed after 16 hr.

## Laser capture microdissection of adrenal cortex

Adrenal glands frozen in liquid nitrogen were cut in 25–30 µm thick sections, mounted on poly-ethylene naphthalate membrane slides (Zeiss), dehydrated in increasing concentrations of ice-cold ethanol (75%, 95%, 100%) for 45 s each, and air-dried at room temperature (RT). Laser capture micro-dissection was performed with a Zeiss PALM MicroBeam LCM system. The adrenal cortex from 8 to 12 sections was microdissected and the tissue was collected on Adhesive Caps (Zeiss).

## Bioinformatic analysis of RNA-Seq data

For transcriptome mapping, strand-specific paired-end sequencing libraries from total RNA were constructed using TruSeq stranded Total RNA kit (Illumina Inc). Sequencing was performed on an Illumina HiSeq3000 (1×75 basepairs). Low-quality nucleotides were removed with the Illumina fastq filter and reads were further subjected to adaptor trimming using cutadapt (*Martin, 2011*). Alignment of the reads to the Mouse Genome was done using STAR Aligner (*Dobin et al., 2013*) using the parameters: '−runMode alignReads −outSAMstrandField intronMotif −outSAMtype BAM SortedByCoordinate --readFilesCommand zcat'. Mouse Genome version GRCm38 (release M12 GENCODE) was used for the alignment. The parameters: 'htseq-count -f bam -s reverse -m union -a 20', HTSeq-0.6.1p1 (*Anders et al., 2015*) were used to count the reads that map to the genes in the aligned sample files. The GTF file (gencode.vM12.annotation.gtf) used for read quantification was downloaded from Gencode (https://www.gencodegenes.org/mouse/release_M12.html). Gene-centric differential expression analysis was performed using DESeq2_1.8.1 (*Anders and Huber, 2010*). The raw read counts for the genes across the samples were normalized using 'rlog' command of DESeq2 and subsequently these values were used to render a PCA plot using ggplot2_1.0.1 (*Wickham, 2009*).

Pathway and functional analyses were performed using GSEA (*Subramanian et al., 2005*) and EGSEA (*Alhamdoosh et al., 2017*). GSEA is a stand-alone software with a GUI. To run GSEA, a ranked

list of all the genes from DESeq2-based calculations was created by taking the -log10 of the p-value and multiplying it with the sign of the fold change. This ranked list was then queried against MSigDB, Reactome, KEGG, and GO-based repositories. EGSEA is an R/Bioconductor-based command-line package. For doing functional analyses using EGSEA, a differentially expressed list of genes with parameters log2fc >0.3 and padj <0.05 was used. Same database repositories as above were used for performing the functional analyses.

For constructing pathway-specific heatmaps, the 'rlog-normalized' expression values of the significantly expressed genes (padj <0.05) were mapped on to the KEGG and GO pathways. These pathway-specific expression matrices were then scaled using Z-transformation. The resulting matrices were visually rendered using MORPHEUS.

## Cell sorting

The adrenal cortex was separated from the medulla under a dissecting microscope and was digested in 1.6 mg/ml collagenase I (Sigma-Aldrich) and 1.6 mg/ml BSA in PBS, for 25 min at 37°C while shaking at 900 rpm. The dissociated tissue was passed through a 22 G needle and 100 µm cell strainer and centrifuged at $300 \times g$ for 5 min at 4°C. The cell suspension was washed in MACS buffer (0.5% BSA, 2 mM EDTA in PBS) and $CD31^+$ and $CD45^+$ cells were sequentially positively selected using anti-CD31 and anti-CD45 MicroBeads (Miltenyi Biotec), respectively, according to the manufacturer's instructions. Briefly, pelleted cells resuspended in 190 µl MACS buffer were mixed with 10 µl anti-CD31 MicroBeads, incubated for 15 min at 4°C, washed with 2 ml MACS buffer, and centrifuged at $300 \times g$ for 10 min at 4°C. Then, the cell pellet was resuspended in 500 µl MACS buffer, applied onto MS Columns placed on MACS Separator, and the flow-through ($CD31^-$ cells) was collected. $CD31^+$ cells were positively sorted from the MS Columns. The flow-through was centrifuged at $300 \times g$ for 5 min at 4°C, and the pelleted cells were subjected to the same procedure using anti-CD45 MicroBeads, collecting the flow-through containing $CD31^-CD45^-$ adrenocortical cells. $CD45^+$ cells were positively sorted from the MS Columns.

## MS/MS proteomic analysis

$CD31^-CD45^-$ adrenocortical cells were sorted and snap-frozen. Samples were randomized and a gel-based sample preparation protocol was followed (*Chen et al., 2011*). In brief, cell pellets were resuspended in SDS loading buffer and 30% acrylamide, boiled at 98°C for 6 min, and 5 µg protein per sample were separated in 10% SDS gels (SurePAGE Bis-Tris gels, GenScript) for approximately 10 min at 120 V. The gels were fixed in 50% (vol/vol) ethanol and 3% (vol/vol) phosphoric acid and briefly stained with Colloidal Coomassie Blue. Sample containing lanes were sliced and cut into blocks of approximately 1 $mm^3$, destained in 50 mM $NH_4HCO_3$ and 50% (vol/vol) acetonitrile, dehydrated using 100% acetonitrile, and rehydrated in 50 mM $NH_4HCO_3$ containing 10 µg/ml trypsin (sequence grade; Promega). After incubation overnight at 37°C peptides were extracted and collected in a new tube, dried using a SpeedVac (Eppendorf), and stored at -20°C until LC-MS analysis. Peptides were dissolved in 0.1% formic acid, and 75 ng were loaded into EvoTips (EV2003, Evosep) and washed according to the manufacturer's guidelines. The samples were run on a 15 cm × 75 µm, 1.9 µm Performance Column (EV1112, Evosep) using the Evosep One liquid chromatography system with the 30 samples per day program. Peptides were analyzed by the TimsTof pro2 mass spectrometer (Bruker) with the diaPASEF method (*Meier et al., 2020*).

Data were analyzed using DIA-NN. The fasta database used was uniprot mouse_UP000000589_10090. Deep learning was used to generate the in silico spectral library. Output was filtered at 0.01 FDR (*Demichev et al., 2020*). The Mass Spectrometry Downstream Analysis Pipeline (MS-DAP) (version beta 0.2.5.1) (https://github.com/ftwkoopmans/msdap) (*Koopmans et al., 2022*; *Koopmans et al., 2023*) was used for quality control and candidate discovery (*Hondius et al., 2021*). Differential abundance analysis between groups was performed on log transformed protein abundances. Empirical Bayes moderated t-statistics with multiple testing correction by FDR, as implemented by the eBayes functions from the limma R package, was used as was previously described (*Koopmans et al., 2018*).

## Bioinformatics analysis of proteomics data

From the proteomics data, the missing data was imputed using the 'impute' command running of 'DEP' (*Zhang et al., 2018*) package in R/Bioconductor (*R Development Core Team, 2018*)

**Table 5.** Primer sequences.

| Gene name | Forward sequence (5' → 3') | Reverse sequence (5' → 3') |
|---|---|---|
| Mouse *18*S rRNA | GTTCCGACCATAAACGATGCC | TGGTGGTGCCCTTCCGTCAAT |
| Mouse *Idh1* | GTGGTGGAGATGCAAGGAGAT | TGGTCATTGGTGGCATCACG |
| Mouse *Idh2* | GATGGACGGTGACGAGATGAC | GGTCTGGTCACGGTTTGGA |
| Mouse *Sdhb* | GGACCTCAGCAAAGTCTCCAA | TGCAGATACTGTTGCTTGCC |
| Mouse *Sdhc* | GCTAAGGAGGAGATGGAGCG | AGAGACCCCTCCACTCAAGG |
| Mouse *Star* | CTGTCCACCACATTGACCTG | CAGCTATGCAGTGGGAGACA |
| Mouse *Cyp11b1* | TCACCATGTGCTGAAATCCTTCCA | GGAAGAGAAGAGAGGGCAATGTGT |
| Mouse *Hsd3b2* | GCGGCTGCTGCACAGGAATAAAG | TCACCAGGCAGCTCCATCCA |
| Mouse *Cyp21a1* | TGGGGATGCAAGATGTGGTGGT | GGTCGGCCAGCAAAGTCCAC |
| Mouse *Cyp11a1* | GGATGCTGGAGGAGATCGT | GAAGTCTGGAGGCAGGTTGA |
| Mouse *Cd31* | TGCAGGAGTCCTTCTCCACT | ACGGTTTGATTCCACTTTGC |
| Mouse *Cd45* | CCAGTCATGCTACCACAACG | TGGACATCTTTGAGGTCTGCC |
| Mouse *Th* | AAGGGCCTCTATGCTACCCA | GCCAGTCCGTTCCTTCAAGA |
| Mouse *Pnmt* | GCATCACATCACCACACTGC | CGGACCTCGTAACCACCAAG |
| Mouse *Acod1* | CTCCCACCGACATATGCTGC | GCTTCCGATAGAGCTGTGA |
| Mouse *Il1r1* | TGGAAGTCTTGTGTGCCCTT | TCCGAAGAAGCTCACGTTGT |
| Mouse *Dnmt1* | CTGGAAGAGGTAACAGCGGG | CGTCCAAGTGAGTTTCCGGT |
| Human *18*S | TGCCCTATCAACTTTCGATG | GATGTGGTAGCCGTTTCTCA |
| Human *SDHB* | CAAGGCTGGAGACAAACCTCA | GGGTGCAAGCTAGAGTGTTG |
| Human *DNMT1* | GGAGGGCTACCTGGCTAAAG | CTGCCATTCCCACTCTACGG |
| Human methylated *SDHB* promoter | AGTGGGTCCTCAGTGGATGTA | GGCGATAGTTTGGTGGCAGA |
| Human unmethylated *SDHB* promoter | CGCGATGTTCGACGGGATA | CTTCACACCCCGCAAATCTC |

environment. The imputation was performed using 'knn' function. The resultant imputed matrix was used for further analyses. Pathway and functional analyses were performed using GSEA (*Subramanian et al., 2005*) and EGSEA (*Alhamdoosh et al., 2017*). CLI version of GSEA v4.1 was run using the imputed matrix. Different pathway sets from MSigDB v7.2 like HALLMARK, Biocarta, Reactome, KEGG, GO, and WIKIPATHWAYS were queried for functional enrichment. Gene set permutations were performed 1000 times to calculate the different statistical parameters. For doing functional analyses using EGSEA, imputed matrix was used. Same database repositories as above were used for performing the functional analyses.

## Quantitative RT-PCR

Total RNA was isolated from frozen adrenal glands with the TRI Reagent (MRC) after mechanical tissue disruption, extracted with chloroform and the NucleoSpin RNA Mini kit (Macherey-Nagel). Total RNA from sorted cells was isolated with the Rneasy Plus Micro Kit (QIAGEN) according to the manufacturer's instructions. cDNA was synthesized with the iScript cDNA Synthesis kit (Bio-Rad) and gene expression was determined using the SsoFast Eva Green Supermix (Bio-Rad), with a CFX384 real-time System C1000 Thermal Cycler (Bio-Rad) and the Bio-Rad CFX Manager 3.1 software. The relative gene expression was calculated using the ΔΔCt method, *18*S was used as a reference gene. Primers are listed in *Table 5*.

## Cell culture and in vitro treatments

CD31⁻CD45⁻ adrenocortical cells were plated on 0.2% gelatin-coated wells of 96-well plates in DMEM/F12 medium supplemented with 1% fetal bovine serum (FBS), 50 U/ml penicillin, and 50 µg/ml streptomycin (all from Gibco), and let to attach for an hour before treatments. Cells from both adrenal glands from each mouse were pooled together and plated in two wells of a 96-well plate. Mouse adrenal explants were dissected from surrounding fat and left in DMEM/F12 medium with 1% FBS, 50 U/ml penicillin, and 50 µg/ml streptomycin for an hour before treatments. NCI-H295R cells (purchased from ATCC) were maintained in DMEM/F12 medium supplemented with 2.5% Nu-Serum type I (Corning), 1% Insulin Transferrin Selenium (ITS; Gibco), 50 U/ml penicillin, and 50 µg/ml streptomycin. NCI-H295R cells were tested mycoplasma-free.

Cells or explants were treated with DMM (20 mM; Sigma-Aldrich), DES (5 mM; Sigma-Aldrich), FCCP (1 µM; Agilent Technologies), OM (500 nM; Agilent Technologies), AG-221 (10 µM; Selleckchem), 4-OI (125 µM; Cayman Chemical), Trolox (20 µM, Abcam), mouse recombinant IL-1β (20 ng/ml, PeproTech), human recombinant IL-1β (20 ng/ml, PeproTech), human recombinant IL-6 (20 ng/ml, PeproTech), human recombinant TNFα (20 ng/ml, PeproTech), LPS (1 µg/ml; InVivoGen), ACTH (100 ng/ml; Sigma-Aldrich), or Forskolin (10 µM; Sigma-Aldrich). siRNA transfections were done with ON-TARGETplus SMARTpool siRNA against *SDHB* (10 nM), *Sdhb* (30 nM), *Idh2* (30 nM), *Dnmt1* (30 nM), or *DNMT1* (30 nM) (all from Horizon Discovery), with Lipofectamine RNAiMAX transfection reagent (Invitrogen), using a reverse transfection protocol per manufacturer's instructions.

## Steroid hormone measurement

Steroid hormones were analyzed by LC-MS/MS in cell culture or explant supernatants as described previously (*Peitzsch et al., 2015*). Fifty to hundred µL cell culture supernatants were extracted by solid phase extraction using positive pressure, followed by a dry-down under gentle stream of nitrogen. Residues were reconstituted in 100 µl of the initial LC mobile phase and 10 µl were injected for detection by the triple quadrupole mass spectrometer in multiple reaction-monitoring scan mode using positive atmospheric pressure chemical ionization. Quantification of steroid concentrations was done by comparisons of ratios of analyte peak areas to respective peak areas of stable isotope labeled internal standards obtained in samples to those of calibrators.

## Measurement of TCA cycle metabolites

TCA cycle metabolites were determined by LC-MS/MS as described before (*Richter et al., 2019*). Itaconate was included in the existing LC-MS/MS method using multi-reaction monitoring (MRM)-derived ion transition of 128.9→85.1. For quantification of itaconate ratios of analyte peak areas to respective peak areas of the stable isotope labeled internal standard (itaconic acid-[13]C5; Bio-Connect B.V., The Netherlands; MRM transition 133.9→89.1) obtained in samples were compared to those of calibrators.

## MALDI-FT-ICR-MSI

Tissue preparation steps for MALDI-MSI analysis was performed as previously described (*Aichler et al., 2017*; *Sun et al., 2018*). Frozen mouse adrenals were cryosectioned at 12 µm (CM1950, Leica Microsystems, Wetzlar, Germany) and thaw-mounted onto indium-tin-oxide-coated conductive slides (Bruker Daltonik, Bremen, Germany). The matrix solution consisted of 10 mg/ml 1,5-diaminonaphthalene (Sigma-Aldrich, Germany) in water/acetonitrile 30:70 (vol/vol). SunCollect automatic sprayer (Sunchrom, Friedrichsdorf, Germany) was used for matrix application. The MALDI-MSI measurement was performed on a Bruker Solarix 7T FT-ICR-MS (Bruker Daltonik, Bremen, Germany) in negative ion mode using 100 laser shots at a frequency of 1000 Hz. The MALDI-MSI data were acquired over a mass range of m/z 75–250 with 50 µm lateral resolution. Following the MALDI imaging experiments, the tissue sections were stained with hematoxylin and eosin and scanned with an AxioScan.Z1 digital slide scanner (Zeiss, Jena, Germany) equipped with a ×20 magnification objective. After the MALDI-MSI measurement, the acquired data underwent spectra processing in FlexImaging v. 5.0 (Bruker Daltonik, Bremen, Germany) and SciLS Lab v. 2021 (Bruker Daltonik, Bremen, Germany). The mass spectra were root-mean-square normalized. MS peak intensity of isocitrate and succinate of adrenal cortex regions were exported and applied for relative quantification analysis.

## ATP measurement

Total ATP was measured in adrenal glands using the ATP Assay Kit (ab83355, Abcam). Briefly, adrenal glands were collected, washed with PBS, and immediately homogenized in 100 µl ATP assay buffer. Samples were cleared using the Deproteinizing Sample Preparation Kit – TCA (ab204708, Abcam). Samples were incubated for 30 min with the ATP reaction mix and fluorescence (Ex/Em = 535/587 nm) was measured using the Synergy HT microplate reader. The recorded measurements were normalized to the weight of the adrenal gland.

## ROS measurement

ROS was detected using the DCFDA/H2DCFDA Cellular ROS Detection Assay Kit (ab113851, Abcam). NCI-H295R cells were plated at 80,000 cells/well in 96-well plate with black walls and clear bottom (Corning) and were incubated with 20 µM DCFDA Solution for 45 min at 37°C in dark. Fluorescence (Ex/Em = 485/535 nm) was measured using the Synergy HT microplate reader.

## ATP/ADP ratio measurement

Intracellular ATP/ADP ratio was determined with the ADP/ATP Ratio Assay Kit (MAK135, Sigma-Aldrich). NCI-H295R cells were plated at 80,000 cells/well in 96-well plate with white flat-bottom wells (Corning). Luminescence was measured using the Synergy HT microplate reader.

## NADPH/NADP$^+$ and NADPH measurement

Intracellular NADPH/NADP$^+$ ratio was measured with the NADP/NADPH Assay Kit (Fluorometric) (ab176724, Abcam). NCI-H295R cells were plated at $5\times10^6$ cells/10 cm – diameter dish. Fluorescence (Ex/Em = 540/590 nm) was measured using the Synergy HT microplate reader. NADPH levels in adrenal tissue homogenates were analyzed by LC-MS/MS using an adapted method as previously described (*Yuan et al., 2012*).

## Enzyme activity measurement

SDH and IDH activities were measured using respective colorimetric assay kits (MAK197, Sigma-Aldrich, ab102528, Abcam). Cortices from both adrenal glands of each mouse were pooled and processed together. Absorbance (at 600 nm for SDH or 450 nm for IDH) was detected using the Synergy HT microplate reader.

## Seahorse assay

OCR measurements were performed with a Seahorse XF96 Analyzer (Agilent Technologies). NCI-H295R cells were plated at 80,000 cells/well in 0.2% gelatin-precoated XF96 cell culture microplate (Agilent). The experimental medium used was XF Base Medium supplemented with glucose (10 mM), pyruvate (1 mM), and glutamine (2 mM).

## Measurement of mitochondrial load and membrane potential

The adrenal cortex was digested and dissociated cells were incubated with Mitotracker Green (0.25 µM; Thermo Fisher), TMRE (2.5 µM; Thermo Fisher), CD31-PeCy7 (1:100; eBioscience), and CD45-PeCy7 (1:100; eBioscience) for 30 min in FACS buffer (0.5% BSA, 2 mM EDTA in PBS) at 37°C in dark. Live cells were selected by Hoechst staining. NCI-H295R cells were incubated with Mitotracker Green (100 nM) and TMRE (100 nM) for 30 min at 37°C in dark. FACS was performed using LSR Fortessa X20 flow cytometer and data were analyzed with the FlowJo software.

## Western blotting

Cells were lysed with 10 mM Tris-HCl, pH7.4+1% SDS+1 mM sodium vanadate, cell lysates were centrifuged at 16,000 × *g* for 5 min at 4°C, supernatants were collected and total protein concentration was measured using Pierce BCA Protein Assay Kit (Thermo Scientific). Gel electrophoresis was performed according to standard protocols (*Laemmli, 1970*). Protein samples were prepared with 5× Reducing Laemmli buffer, denatured at 95°C for 5 min and loaded on a 10% acrylamide gel (Invitrogen) for sodium dodecyl sulfate polyacrylamide gel electrophoresis. PageRuler Prestained Protein Ladder (Thermo Fisher Scientific) was used as a protein size ladder. The separated proteins were transferred on Amersham Protran nitrocellulose membrane (GE Healthcare Lifescience). After

blocking with 5% skimmed milk in TBS-T (0.1% Tween-20 [Sigma-Aldrich] in 1× Tris-buffered saline) for 1 hr at RT, membranes were incubated overnight at 4°C with anti-SDHB (1:1000; Sigma-Aldrich, HPA002868), anti-DNMT1 (1:1000; Cell Signaling, #5032), anti-Tubulin (1:3000; Sigma-Aldrich, T5186), or anti-β-Actin (1:1000; Cell Signaling, #4967), diluted in 5% BSA in TBS-T. After washing, membranes were incubated for 1 hr at RT with secondary antibodies: goat anti-rabbit IgG HRP-conjugated (1:3000; Jackson ImmunoResearch) or goat anti-mouse IgG HRP-conjugated (1:3000; Jackson ImmunoResearch), diluted in 5% skimmed milk in TBS-T. The signal was detected using the Western Blot Ultra-Sensitive HRP Substrate (Takara) and imaged using the Fusion FX Imaging system (PeqLab Biotechnologie).

## DNA methylation measurement

Genomic DNA from $2×10^6$ NCI-H295R cells was isolated with the Quick-DNA Miniprep Kit (Zymo Research). Bisulfite treatment was performed using the EZ DNA Methylation Kit (Zymo Research), following the manufacturer's protocol. For each sample, 500 ng genomic DNA was used, bisulfite treated for 14 hr in the dark and, after a desulphonation and cleaning step, eluted in 10 µl nuclease-free water. The SDHB promoter region was amplified with primers for a methylated and a non-methylated sequence (listed in *Table 5*), using the QIAGEN Multiplex PCR Kit. Equal amount of DNA not treated with bisulfite was amplified as a loading control. The PCR products were then electrophoresed on 3% agarose gel and visualized under UV illumination using the Fusion FX Imaging system (Vilber). The ratio of methylated to non-methylated DNA was calculated after gel intensity quantification in ImageJ.

## Immunofluorescent staining

Adrenal glands cleaned from surrounding fat tissue were fixed in 4% PFA in PBS, washed overnight in PBS, cryopreserved in 30% sucrose (AppliChem GmbH) in PBS overnight at 4°C, embedded in OCT compound (Tissue-Tek), and frozen at −80°C. Each adrenal gland was cut into 8 µm thick serial sections. Before staining, adrenal sections were pre-warmed at RT for 30 min and antigen retrieval was performed by boiling in citrate buffer (pH 6) for 6 min. Adrenal sections were washed with PBS, permeabilized with 0.1% Triton X-100 in PBS for 20 min, treated with TrueBlack Lipofuscin Quencher (1:40 in 70% ethanol; Biotium) for 30 s to reduce autofluorescence and blocked in Dako Protein Block, serum-free buffer for 1 hr at RT. Then, sections were incubated overnight at 4°C with primary antibodies, washed with PBS, and incubated for 1 hr at RT with the secondary antibodies together with DAPI (1:5000; Roche), all diluted in Dako Antibody Diluent. Antibodies and dyes used were: anti-SDHB (1:300; Sigma-Aldrich, HPA002868), anti-IDH2 (1:50; Sigma-Aldrich, HPA007831), anti-SF-1 (1:100; TransGenic Inc KO610), Lectin Esculentum DyLight488 (1:300; Vector Laboratories, DL-1174), 4-HNE (1:200; Abcam, ab48506), Alexa Fluor 555 donkey anti-rabbit (1:300; Life Technologies, #A-31572), Alexa Fluor 647 chicken anti-rat (1:300; Invitrogen, #A21472), and Alexa Fluor 555 donkey anti-mouse (1:300; Invitrogen, #A31570). After washing with PBS, cryosections were mounted with Fluoromount (Sigma-Aldrich), covered with 0.17 mm cover glass, fixed with nail polish, and kept at 4°C until imaging.

## Image acquisition and image analysis

Z-series microscopic images for SDHB and IDH2 staining were acquired on Zeiss LSM 880 inverted confocal microscope (Zeiss, Jena, Germany), illuminated with laser lines at 405 nm, 488 nm, 561 nm, and 633 nm, and detected by two photomultiplier tube detectors. EC Plan-Neofluoar objective with ×40 magnification, 1.30 numerical aperture, and M27 thread, working with an oil immersion medium Immersol 518F, was used. Microscopic images of SF-1 and 4-HNE stainings were acquired with an Axio Observer Z1/7 inverted microscope with Apotome mode (Zeiss, Jena, Germany), illuminated with LED-Module 385 nm and 567 nm, on a Plan-Apochromat objective with ×10 magnification, 0.45 numerical aperture, and M27 thread. Laser power, photomultiplier gain, and pinhole size were set for each antibody individually and kept constant for all image acquisitions. For each condition, at least three view-fields were imaged per tissue section. Images were acquired with the ZEN 3.2 blue edition software, and processed and quantified with the ImageJ software on maximum intensity Z-projection images.

## Statistical analysis

The statistical analysis and data plotting were done with the GraphPad Prism 7.04 software. The statistical tests used are described in each figure legend, $p < 0.05$ was set as a significance level.

## Graphical design

*Figure 7* was created with Biorender.com.

## Acknowledgements

We thank Christine Mund, Denise Kaden, and Catleen Conrad for technical assistance. We acknowledge the technical support from the Core Facility Cellular Imaging of the Medical Faculty Carl Gustav Carus, TU Dresden for confocal imaging, and from the Light Microscopy Facility of the CMCB Technology Platform at TU Dresden, for laser microdissection.

## Additional information

### Funding

| Funder | Grant reference number | Author |
| --- | --- | --- |
| Deutsche Forschungsgemeinschaft | SFB/TRR205 | Ben Wielockx Mirko Peitzsch Vasileia Ismini Alexaki |
| HORIZON EUROPE Framework Programme | Marie Skłodowska-Curie grant agreement No 765704 | Vasileia Ismini Alexaki |

The funders had no role in study design, data collection and interpretation, or the decision to submit the work for publication.

### Author contributions

Ivona Mateska, Data curation, Formal analysis, Validation, Investigation, Visualization, Methodology, Writing - original draft; Anke Witt, Eman Hagag, Validation, Investigation, Methodology; Anupam Sinha, Data curation, Software, Formal analysis; Canelif Yilmaz, Investigation, Methodology; Evangelia Thanou, Data curation, Software, Formal analysis, Validation, Investigation; Na Sun, Formal analysis, Validation, Investigation, Methodology; Ourania Kolliniati, Maria Patschin, Heba Abdelmegeed, Waldemar Kanczkowski, Investigation; Holger Henneicke, Ben Wielockx, Writing - review and editing; Christos Tsatsanis, Axel Karl Walch, Ka Wan Li, Triantafyllos Chavakis, Resources; Andreas Dahl, Software, Formal analysis; Mirko Peitzsch, Resources, Data curation, Software, Formal analysis, Validation; Vasileia Ismini Alexaki, Conceptualization, Resources, Supervision, Funding acquisition, Visualization, Writing - original draft, Project administration, Writing - review and editing

### Author ORCIDs

Ivona Mateska (iD) http://orcid.org/0000-0001-6150-9175
Canelif Yilmaz (iD) http://orcid.org/0000-0002-9676-9310
Evangelia Thanou (iD) http://orcid.org/0000-0001-6843-4591
Andreas Dahl (iD) http://orcid.org/0000-0002-2668-8371
Ka Wan Li (iD) http://orcid.org/0000-0001-6983-5055
Mirko Peitzsch (iD) http://orcid.org/0000-0002-2472-675X
Vasileia Ismini Alexaki (iD) http://orcid.org/0000-0003-3935-8985

### Ethics

Ethics approvalThe animal experiments were approved by the Landesdirektion Sachsen Germany (protocol number TVV57/2018).

### Decision letter and Author response

Decision letter https://doi.org/10.7554/eLife.83064.sa1
Author response https://doi.org/10.7554/eLife.83064.sa2

## Additional files

**Supplementary files**
• MDAR checklist

**Data availability**

RNA-Seq data are available in: https://www.ncbi.nlm.nih.gov/geo/query/acc.cgi?acc=GSE200220. The mass spectrometry proteomics data have been deposited to the ProteomeXchange Consortium via the PRIDE partner repository, with the dataset identifier PXD036542.

The following datasets were generated:

| Author(s) | Year | Dataset title | Dataset URL | Database and Identifier |
|---|---|---|---|---|
| Alexaki IV, Sinha A | 2022 | Effect of LPS treatment on gene expression in the murine adrenal cortex | https://www.ncbi.nlm.nih.gov/geo/query/acc.cgi?acc=GSE200220 | NCBI Gene Expression Omnibus, GSE200220 |
| Alexaki IV, Thanou E | 2022 | Inflammation-induced metabolic reprogramming of adrenocortical cells regulates steroidogenesis | https://www.ebi.ac.uk/pride/archive/projects/PXD036542 | PRIDE, PXD036542 |

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
