## [Editor Report]

Acute inflammation in mammals activates the hypothalamic pituatary axis leading to increased glucocorticoid release, which is required to restrain the inflammatory response. However, in settings of severe or prolonged inflammation, such as that seen in sepsis, there is reduced adrenal steridogenesis. The studies described in this paper provide a plausible mechanism for adrenal resistance which develops during excessive inflammation. The revisions have improved the paper and the Methods are sound.

---

## [Decision Letter]

**Decision letter after peer review:**

Thank you for submitting your article "Succinate mediates inflammation-induced adrenocortical dysfunction" for consideration by *eLife*. Your article has been reviewed by 3 peer reviewers, including Thomas L Clemens as Reviewing Editor and Reviewer #1, and the evaluation has been overseen by Mone Zaidi as the Senior Editor. The following individuals involved in review of your submission have agreed to reveal their identity: Christian A. Koch (Reviewer #3).

Essential revisions:

1) The initial in vivo findings, which support the proposed metabolic perturbation, are based on descriptive profiling data obtained at one time point following a single dose of LPS. The author's conclusion that the ultimate transcriptional pathway identified hinges critically on knowledge of the time course of this effect following LPS, which is not adequately addressed in the paper. How was this time and dose of LPS established and are there data from different dose and time points?

2) The authors should confirm through direct biochemical assays of enzymatic activity that steroidogenesis enzyme activity is not impaired. Many of these enzymes are located in the mitochondria and their activity may be diminished due to the disturbed, high succinate environment of the cortical cell as opposed to the low ATP production.

3) The proposed connection of DNMT and IL1 signaling to systemic inflammation and reduced steriodogenesis could be more firmly established by additional studies in adrenal cortical cells lacking these genes.

*Reviewer #3 (Recommendations for the authors):*

The authors provide a well-organized and well written study of LPS induced systemic inflammation effects on adrenal steroidogenesis.

Utilizing 8–12-week-old male mice and various analyses at the transcriptome, proteome and metabolome level the authors show that increasing succinate and decreasing ATP levels disrupt adrenal steroid production.

Comments

1. It is great to see that male mice aged 8-12 weeks were used rather than younger ones

Nevertheless, it is typically difficult to achieve 100% purity when dissecting adrenal glands of such mice.

The ZR is absent and there is a prominent x zone at the corticomedullary junction. This may have potentially had an impact on the results

2. Typos on page 3, introduction line 5.

"Formatting citation" is missing the references.

Page 24, reference section.

Ref 1: hypothalamic.

Ref 7: journal Endocrinology.

Please double check all refs.

---

## [Author Response]

Essential revisions:1) The initial in vivo findings, which support the proposed metabolic perturbation, are based on descriptive profiling data obtained at one time point following a single dose of LPS. The author's conclusion that the ultimate transcriptional pathway identified hinges critically on knowledge of the time course of this effect following LPS, which is not adequately addressed in the paper. How was this time and dose of LPS established and are there data from different dose and time points?

We thank the Reviewing Editor for raising this question, which we indeed addressed at the beginning of our studies in order to determine a suitable time point and dose of LPS treatment. We chose 6 h as a suitable starting time point to perform transcriptional analyses, based on the fact that LPS triggers transcriptional changes in the adrenal gland and other tissues within the range of few hours (*1-3*). Confirming our expectations, we found 2,609 differentially expressed genes (Figure 1a) in the adrenal cortex of LPS-treated mice among which many were involved in cellular metabolism (Figure 1d,e, 2a-e, Table 1, Table 2). Acute transcriptional changes, which are more likely to reflect direct effects of inflammatory signals compared to changes occurring at later time points (for instance in the range of days), would allow us to mechanistically investigate the effects of inflammation in the adrenal gland, which was the purpose of our studies. Hence, we were guided by the transcriptional changes observed at 6 h of LPS treatment and established the hypothesis that disruption of the TCA cycle in adrenocortical cells is central in the impact of inflammation on adrenal function. Along this line, we analyzed the metabolomic profile of the adrenal gland at 6 and 24 h of LPS treatment. At 6 h succinate levels as well as the succinate / fumarate ratio remained unchanged (Author response image 1), while at 24 h post-injection these were increased by LPS (Author response image 1; Figure 2l,o,q). The time delay of the increase in succinate levels (observed at 24 h) following downregulation of *Sdhb* mRNA expression (at 6 h) can be explained by the time required for reduction of SDHB protein levels, which is dependent on the protein turnover suggested to be approximately 12 h in HeLa cells (*4*). Based on these findings, all further metabolomic analyses were performed at 24 h of LPS treatment.

**Author response image 1. sa2fig1:** LPS increases the succinate/fumarate ratio at 24 but not 6 h. Mice were i.p. injected with 1 mg/kg LPS and 6 h (A) and 24 h (B) post-injection succinate and fumarate levels were determined by LC-MS/MS in the adrenal gland. n=8-10; data are presented as mean ± s.e.m. Statistical analysis was done with two-tailed Mann-Whitney test. *p < 0.05.

Having established the most suitable time points of LPS treatments to observe induced transcriptional and metabolic changes, we set out to define the LPS dose to be used in subsequent experiments. The data shown in Author response image 1, were acquired after treatment with 1 mg/kg LPS. This is a dose that was previously reported to cause transcriptional re-profiling of the adrenal gland (*1, 2*). However, 5 mg/kg LPS, similarly to 1 mg/kg LPS, also reduced *Sdhb*, *Idh1* and *Idh2* expression at 4 h (Author response image 2) and increased succinate and isocitrate levels at 24 h (Author response image 2) in the adrenal gland. Given that the effects of 1 and 5 mg/kg LPS were similar, for animal welfare reasons we continued our studies with the lower dose.

**Author response image 2. sa2fig2:** Five mg/kg LPS downregulate *Sdhb*, *Idh1* and *Idh2* expression and increase succinate and isocitrate levels in the adrenal gland of mice. *Sdhb*, *Idh1* and *Idh2* expression (A) and succinate and isocitrate levels (B) were assessed in the adrenal gland of mice treated with 5 mg/kg LPS for 4 h (A) and 24 h (B). n=5; data are presented as mean ± s.d. Statistical analysis was done with two-tailed Mann-Whitney test. *p < 0.05, **p < 0.01.

2) The authors should confirm through direct biochemical assays of enzymatic activity that steroidogenesis enzyme activity is not impaired. Many of these enzymes are located in the mitochondria and their activity may be diminished due to the disturbed, high succinate environment of the cortical cell as opposed to the low ATP production.

We thank the Reviewer and Reviewing Editor for this excellent question. The activity of the first and rate-limiting steroidogenic enzyme, cytochrome P450-side-chain-cleavage (SCC, CYP11A1) which generates pregnenolone from cholesterol, was recently shown to require intact SDH function (*5*). In agreement with this report we show that production of progesterone, the direct derivative of pregnenolone, is impaired upon SDH inhibition (Figure 5b,e,h). In addition, we assessed the activity of CYP11B1 (steroid 11β-hydroxylase), the enzyme catalyzing the conversion of 11-deoxycorticosterone to corticosterone, i.e. the last step of glucocorticoid synthesis, by determining the corticosterone and 11-deoxycorticosterone levels by LC-MS/MS and calculating the ratio of corticosterone to 11-deoxycorticosterone in ACTH-stimulated adrenocortical cells and explants. The corticosterone / 11-deoxycorticosterone ratio was not affected by *Sdhb* silencing in adrenocortical cells (Figure 5- Supplement 1g) nor did it change upon LPS treatment in adrenal explants (Figure 5- Supplement 1h), suggesting that CYP11B1 activity is probably not altered upon SDH blockage. Hence, we propose that upon inflammation impairment of SDH function may disrupt at least the first steps of steroidogenesis generating pregnenolone and/or progesterone, thereby diminishing production of all downstream adrenocortical steroids. This is now highlighted in the Discussion of our revised manuscript.

3) The proposed connection of DNMT and IL1 signaling to systemic inflammation and reduced steriodogenesis could be more firmly established by additional studies in adrenal cortical cells lacking these genes.

We strengthened the evidence for an IL-1β – DNMT1 link with a series of additional experiments, and show that DNMT1 deficiency blocks the effects of IL-1β on *SDHB* promoter methylation (Figure 6k), the succinate / fumarate ratio (Figure 6m), the oxygen consumption rate (Figure 6n) and steroidogenesis (Figure 6o-q) in adrenocortical cells. In order to validate the role of IL-1β in vivo, mice were simultaneously treated with LPS and Raleukin, an IL-1R antagonist. Treatment with Raleukin increased the SDH activity (Figure 6r), reduced succinate levels and the succinate / fumarate ratio (Figure 6s,t) and increased corticosterone production in LPS-treated mice (Figure 6u).

Reviewer #3 (Recommendations for the authors):The authors provide a well-organized and well written study of LPS induced systemic inflammation effects on adrenal steroidogenesis.Utilizing 8–12-week-old male mice and various analyses at the transcriptome, proteome and metabolome level the authors show that increasing succinate and decreasing ATP levels disrupt adrenal steroid production.Comments1. It is great to see that male mice aged 8-12 weeks were used rather than younger onesNevertheless, it is typically difficult to achieve 100% purity when dissecting adrenal glands of such mice.The ZR is absent and there is a prominent x zone at the corticomedullary junction. This may have potentially had an impact on the results

We thank the Reviewer for this comment. Absence of medullar material in the isolated cortices and adrenocortical cells was verified by the absence of expression of the medullar markers *Th* and *Pnmt1* (Figure 1 —figure supplement 1d,e).

The X-zone, a cluster of eosinophilic cells in the inner cortex next to the medulla with unknown function, is a remnant of the fetal adrenal cortex that regresses at puberty (before P40) in male mice but is retained in females until gestation (*11*). Given that we used 8-12 week old, i.e. adult male mice, we can exclude a potential impact of the X-zone on our findings.

2. Typos on page 3, introduction line 5."Formatting citation" is missing the references.Page 24, reference section.Ref 1: hypothalamic.Ref 7: journal Endocrinology.Please double check all refs.

We thank the Reviewer for carefully reading our manuscript. We corrected the points above and can confirm that all references in the revised manuscript are correct.

References

1. W. Kanczkowski, A. Chatzigeorgiou, M. Samus, N. Tran, K. Zacharowski, T. Chavakis, S. R. Bornstein, Characterization of the LPS-induced inflammation of the adrenal gland in mice. *Mol Cell Endocrinol* 371, 228-235 (2013).

2. L. S. Chen, S. P. Singh, M. Schuster, T. Grinenko, S. R. Bornstein, W. Kanczkowski, RNA-seq analysis of LPS-induced transcriptional changes and its possible implications for the adrenal gland dysregulation during sepsis. *J Steroid Biochem Mol Biol* 191, 105360 (2019).

3. V. I. Alexaki, G. Fodelianaki, A. Neuwirth, C. Mund, A. Kourgiantaki, E. Ieronimaki, K. Lyroni, M. Troullinaki, C. Fujii, W. Kanczkowski, A. Ziogas, M. Peitzsch, S. Grossklaus, B. Sonnichsen, A. Gravanis, S. R. Bornstein, I. Charalampopoulos, C. Tsatsanis, T. Chavakis, DHEA inhibits acute microglia-mediated inflammation through activation of the TrkA-Akt_1/2_-CREB-Jmjd3 pathway. *Mol Psychiatry* 23, 1410-1420 (2018).

4. C. Yang, J. C. Matro, K. M. Huntoon, D. Y. Ye, T. T. Huynh, S. M. Fliedner, J. Breza, Z. Zhuang, K. Pacak, Missense mutations in the human SDHB gene increase protein degradation without altering intrinsic enzymatic function. *FASEB J* 26, 4506-4516 (2012).

5. H. S. Bose, B. Marshall, D. K. Debnath, E. W. Perry, R. M. Whittal, Electron Transport Chain Complex II Regulates Steroid Metabolism. *iScience* 23, 101295 (2020).

6. W. Kanczkowski, V. I. Alexaki, N. Tran, S. Grossklaus, K. Zacharowski, A. Martinez, P. Popovics, N. L. Block, T. Chavakis, A. V. Schally, S. R. Bornstein, Hypothalamo-pituitary and immune-dependent adrenal regulation during systemic inflammation. *Proc Natl Acad Sci U S A* 110, 14801-14806 (2013).

7. W. Kanczkowski, A. Chatzigeorgiou, S. Grossklaus, D. Sprott, S. R. Bornstein, T. Chavakis, Role of the endothelial-derived endogenous anti-inflammatory factor Del-1 in inflammation-mediated adrenal gland dysfunction. *Endocrinology* 154, 1181-1189 (2013).

8. C. Jennewein, N. Tran, W. Kanczkowski, L. Heerdegen, A. Kantharajah, S. Drose, S. Bornstein, B. Scheller, K. Zacharowski, Mortality of Septic Mice Strongly Correlates With Adrenal Gland Inflammation. *Crit Care Med* 44, e190-199 (2016).

9. D. Annane, V. Sebille, G. Troche, J. C. Raphael, P. Gajdos, E. Bellissant, A 3-level prognostic classification in septic shock based on cortisol levels and cortisol response to corticotropin. *JAMA* 283, 1038-1045 (2000).

10. E. Boonen, S. R. Bornstein, G. Van den Berghe, New insights into the controversy of adrenal function during critical illness. *Lancet Diabetes Endocrinol* 3, 805-815 (2015).

11. C. C. Huang, Y. Kang, The transient cortical zone in the adrenal gland: the mystery of the adrenal X-zone. *J Endocrinol* 241, R51-R63 (2019).